# Evolutionary ecology of microbial populations inhabiting deep sea sediments associated with cold seeps

Xiyang Dong [1,2,10] ✉, Yongyi Peng [1,3,10], Muhua Wang[2,3], Laura Woods[4], Wenxue Wu [2,3,5], Yong Wang [6], Xi Xiao[7], Jiwei Li[8], Kuntong Jia[3], Chris Greening [4], Zongze Shao [1,2] ✉ & Casey R. J. Hubert [9]

Deep sea cold seep sediments host abundant and diverse microbial populations that significantly influence biogeochemical cycles. While numerous studies have revealed their community structure and functional capabilities, little is known about genetic heterogeneity within species. Here, we examine intraspecies diversity patterns of 39 abundant species identified in sediment layers down to 430 cm below the sea floor across six cold seep sites. These populations are grouped as aerobic methane-oxidizing bacteria, anaerobic methanotrophic archaea and sulfate-reducing bacteria. Different evolutionary trajectories are observed at the genomic level among these physiologically and phylogenetically diverse populations, with generally low rates of homologous recombination and strong purifying selection. Functional genes related to methane (*pmoA* and *mcrA*) and sulfate (*dsrA*) metabolisms are under strong purifying selection in most species investigated. These genes differ in evolutionary trajectories across phylogenetic clades but are functionally conserved across sites. Intrapopulation diversification of genomes and their *mcrA* and *dsrA* genes is depth-dependent and subject to different selection pressure throughout the sediment column redox zones at different sites. These results highlight the interplay between ecological processes and the evolution of key bacteria and archaea in deep sea cold seep extreme environments, shedding light on microbial adaptation in the subseafloor biosphere.

Cold seeps are widely distributed along continental margins across the globe and serve as highly productive hotspots. Microbial communities in such seeps can be several orders of magnitude more abundant than those in the surrounding marine sediments, potentially impacting how microbial populations evolve over time[1–5]. Cold seeps also possess unique environmental conditions compared to the other benthic environments, shaping local ecology, evolutionary biology, chemistry, and geology[4,6–8]. Seep fluids are enriched in hydrocarbon gases (mainly methane) and other energy-rich hydrocarbon fluids, and markedly alter the sedimentary microbial community structure and function by

[1]Key Laboratory of Marine Genetic Resources, Third Institute of Oceanography, Ministry of Natural Resources, Xiamen 361005, China. [2]Southern Marine Science and Engineering Guangdong Laboratory (Zhuhai), Zhuhai 519000, China. [3]School of Marine Sciences, Sun Yat-Sen University, Zhuhai 519082, China. [4]Department of Microbiology, Biomedicine Discovery Institute, Monash University, Clayton, VIC 3800, Australia. [5]State Key Laboratory of Marine Resource Utilization in South China Sea, Hainan University, Haikou 570228, China. [6]Institute for Ocean Engineering, Shenzhen International Graduate School, Tsinghua University, Shenzhen 518055, China. [7]Guangzhou Marine Geological Survey, China Geological Survey, Guangzhou 510075, China. [8]Institute of Deep-Sea Science and Engineering, Chinese Academy of Sciences, Sanya 572000, China. [9]Department of Biological Sciences, University of Calgary, Calgary, AB T2N 1N4, Canada. [10]These authors contributed equally: Xiyang Dong, Yongyi Peng. ✉e-mail: dongxiyang@tio.org.cn; shaozz@163.com

promoting microbial growth, specialization, and adaptation[4]. Based on amplicon sequencing of genetic markers and genome-resolved metagenomics, numerous studies have revealed the extensive macrodiversity (i.e., the measure of population diversity within a community)[9] of archaeal and bacterial lineages and microbial metabolic versatility along different cold seep sites and sediment depths[3,4,10–12]. Findings include the discovery of various lineages of archaeal anaerobic methanotrophs (ANME) and sulfate-reducing bacteria (SRB) as syntrophic aggregates that perform methane oxidation coupled to sulfate reduction in anoxic sediment layers[13,14]. In the upper oxic sediment layers of cold seeps, methane is consumed by aerobic methane-oxidizing bacteria (MOB), mainly from the order *Methylococcales*[5,12]. Although macro-variations in microbial diversity and functions have been well-characterized, our knowledge of the microdiversity (i.e., the measure of genetic variation within a population)[1,15,16] of these metabolically and taxonomically diverse subseafloor microorganisms remains limited. By addressing crucial questions such as "what populations have high microdiversity levels", "which genes are under selection" and "does homologous recombination occur", intrapopulation microdiversity analyses can provide a complete understanding of microbial ecology and evolution in the subseafloor biosphere[17,18].

Traditional cultivation-based approaches have a fundamental role in studying genetic variation in microbial populations but are often not applicable to the study of the subseafloor biosphere, where most microorganisms are extremely difficult to isolate[15,19,20]. With the emergence of bioinformatic tools for culture-free, high-resolution strain and subspecies analyses in complex environments[9,16,21–23], genome-resolved metagenomic analyses at large-scale can now be conducted to reveal fine-scale evolutionary mechanism dynamics and strain-level metabolic variation[1,15,17,18,24–26]. Pioneering studies have used metagenomic data to explore the roles of basic processes (natural selection, mutation, genetic drift, and recombination) in shaping the microbial evolution of several typical subseafloor habitats. For instance, in situ work examining the genomic variation of microbes inhabiting the upper two meters of anoxic subseafloor sediments in Aarhus Bay revealed that rates of genomic diversification and selection did not change with either sediment age or depth, likely due to energy limitation and reduced growth in this environment[27]. In contrast to non-vent sediments, population-specific differences in selection pressure were observed in both *Sulfurovum* and *Methanothermococcus* species between two geochemically distinct hydrothermal vent fields, where energy availability and cell abundances are relatively high[28,29]. A third study found that gene flow and recombination appeared to shape the evolution of microbial metapopulations that disperse frequently through the cold, oxic crustal fluids of the mid-Atlantic ridge[30]. Cold seep sediments feature a supply of hydrocarbon-rich energy and carbon sources, strong redox zonation over depth, abundant electron donors (e.g. sulfide and methane), and frequent fluid exchange[5,6,12], enabling potential unusual microbial evolution in these environments. Methane- and sulfur-cycling bacteria and archaea are key players in the microbial communities inhabiting the cold seep sediment ecosystems[4,6,11,31]. These populations impose strong controls on local chemical and biological regimes through a variety of biogeochemical processes and interactions[12]. Despite these dynamics, no direct studies of the evolutionary histories and selection pressures of cold seep sedimentary microorganisms have been conducted, and knowledge of the nucleotide variation of key functional genes related to methane- and sulfur-cycling is lacking.

Here, we hypothesize that microbial evolution in cold seep sediments differs from both highly productive deep-sea hydrothermal vents and energy-limited marine sediments, due to the continuous flow of hydrocarbon-rich fluids. To gain insights into evolutionary trajectories among microbial populations inhabiting cold seep sediments, we examine the metagenomic data of 68 cold seep sediment samples to track population microdiversity from metagenomic short-read alignments and perform microdiversity-oriented genomic comparisons. Our study also reveals the depth- and site-dependent trends of microbial evolution in deep-sea cold seep sediments by analyzing the inter-sample genomic variation of species. Microbial adaptation in a gradient environment is a highly dynamic and complex process involving the interaction of multiple evolutionary forces[23]. Thus, the exploration of adaptive fingerprints to uncover evolutionary mechanisms of specific taxa in cold seep sediments may hint at the factors impacting long-term evolution in the deep subseafloor biosphere, as well as those processes that shape the evolution of genes involved in adaptation to specific environmental factors.

## Results

### Depth distribution of species-level clusters in cold seep sediments

We assembled metagenomic data sequenced from 68 sediment samples obtained from six globally distributed cold seep sites. Sediment samples span different depths and redox conditions, from the oxic sediment-water interface into anoxic layers down to 430 cm below the sea floor (cmbsf) (Supplementary Fig. 1 and Supplementary Data 1). After binning of metagenomic assemblies and dereplication of metagenome-assembled genomes (MAGs), 1261 species-level clusters (1041 bacteria and 220 archaea; Supplementary Data 2) were recovered according to the suggested threshold of 95% average nucleotide identity (ANI) for delineating species[32–34]. The 1261 species clusters belonged to 85 phyla (70 bacterial and 15 archaeal; Fig. 1) based on the Genome Taxonomy Database (GTDB; version R06-202)[35–38], and were highly represented by bacteria from the Chloroflexota (*n* = 184), Proteobacteria (*n* = 125), Desulfobacterota (*n* = 101), Planctomycetota (*n* = 73) and Bacteroidota (*n* = 67). The top five archaeal phyla were Asgardarchaeota (*n* = 50), Thermoplasmatota (*n* = 44), Halobacteriota (*n* = 42), Thermoproteota (*n* = 35; mainly *Bathyarchaeia*), and Nanoarchaeota (*n* = 19). The majority of the 85 phyla recovered lack an available cultured representative in the GTDB[37] and consist exclusively of MAGs derived from community DNA and/or single cell-amplified genomes (SAGs) derived from cell sorting of environmental samples. Approximately 51% and 94% of recovered genome clusters could not be assigned to an existing genus or species, respectively (Supplementary Fig. 2), confirming that cold seep microbiomes consist of predominantly uncultured members.

At the phylum level, the total relative abundance of members of the archaeal phylum Halobacteriota was the highest across samples in most sediment depths (Supplementary Fig. 3), and ranged from 2.9 ± 1.2% at the surface (0–5 cmbsf; *n* = 10 samples) to 20.7 ± 7.3% in deeper layers (200–300 cmbsf; *n* = 4 samples) (Supplementary Data 3). At the species level (Supplementary Fig. 4), clusters from ANME-1 (genus QEXZ01) ranked the highest (up to 9.6 ± 2.8% at four depth categories ranging from 30 to 430 cmbsf), followed by ANME-2c groups (up to 2.0 ± 0.4% at four depth categories ranging from 0 to 30 cmbsf), indicating differential depth distributions among ANME archaea[39]. Bacterial taxa belonging to the phyla Desulfobacterota and Caldatribacteriota were also very abundant, with the latter being especially prevalent in deeper sediments (Supplementary Fig. 3). The most abundant bacterial MAG at the species level was Caldatribacteriota SB_S5_bin2, with values of up to 2.2 ± 1.3% at 100–200 cmbsf (Supplementary Fig. 5). Three bacterial species belonging to Desulfobacterota (SB_S3_bin18, S11_7_23, SF_GA_5_3), the prospective syntrophic partners of ANME-1 or ANME-2[40], were relatively predominant (up to 2.3 ± 0.5% at two depth categories ranging from 200 to 450 cmbsf) (Supplementary Fig. 5). This is in agreement with the recent demonstration that depth-dependent distribution patterns were observed for cold seep microbial communities based on 16S rRNA gene and metagenomic sequencing[3]. Overall, sediment depth is one

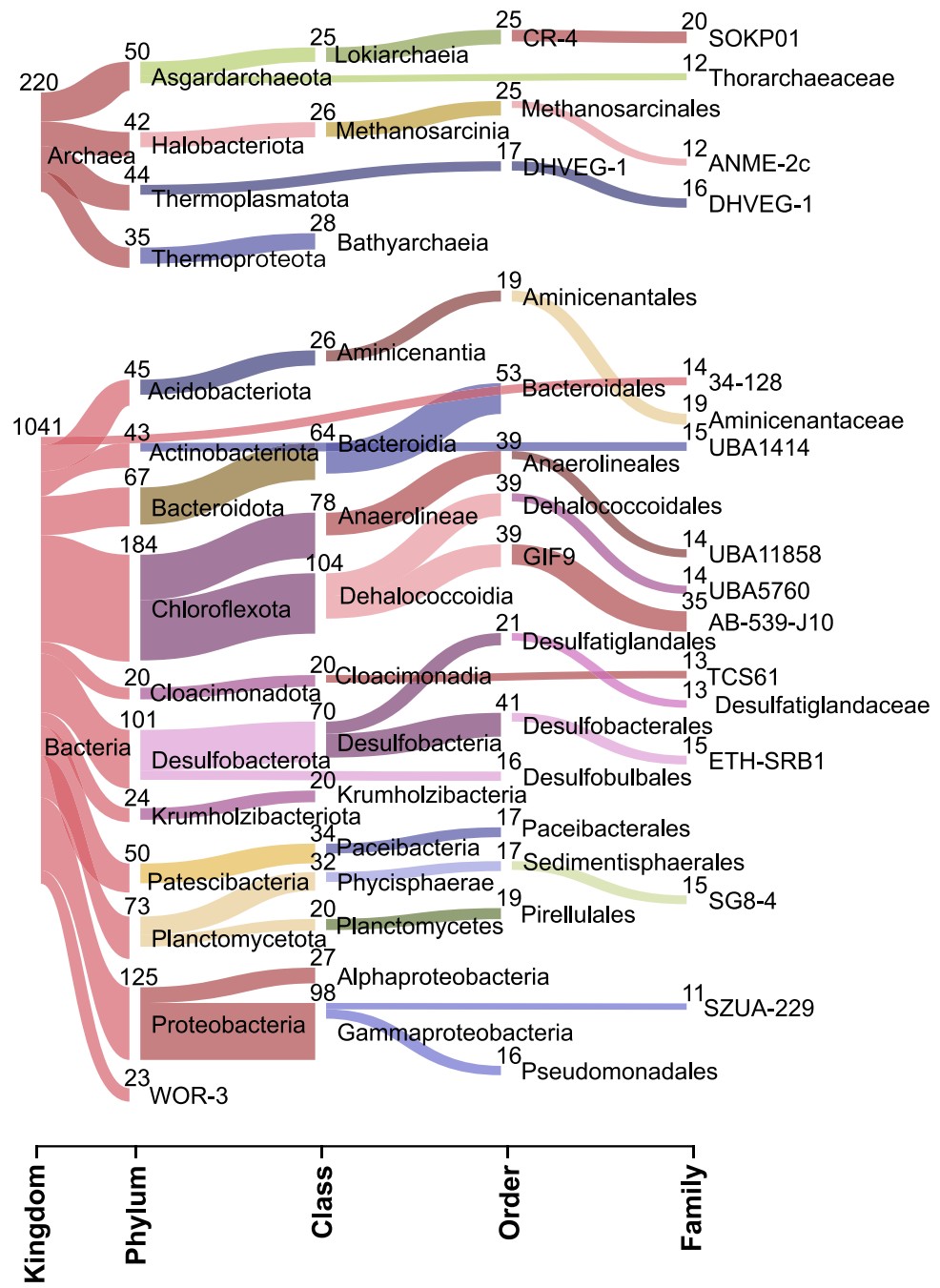

**Fig. 1 | Classification of species-level genomes recovered from global cold seep sediment metagenomes.** Sankey diagram based on assigned GTDB taxonomy showing recovered archaeal and bacterial MAGs at different taxonomic levels, indicating the number of MAGs recovered for a given lineage. The diagram shows the top 25 taxa with the largest number of MAGs at each level. Detailed statistics for 1261 MAGs are provided in Supplementary Data 2.

important factor shaping distributions of cold seep sediment microbial species clusters[41].

## Selecting different species-level genomes for microdiversity analysis

Aerobic methane-oxidizing bacteria (MOB), anaerobic methane-oxidizing (ANME) archaea, and sulfate-reducing bacteria (SRB) are three guilds carrying out fundamental biogeochemical functions in cold seep sediments[4,10,12]. To identify MOB, ANME, and SRB, 1261 MAGs were screened for diagnostic functional genes (Fig. 2): the particulate methane monooxygenase marker gene *pmoA* in aerobic methanotrophic bacteria, the methyl-coenzyme-M reductase marker gene *mcrA* in anaerobic methanotrophic archaea, and the dissimilatory sulfite

reductase marker gene *dsrA* in sulfate reducers. A total of 39 MOB, ANME, and SRB MAGs were retained as species-cluster representatives for microdiversity analyses, which satisfied threshold criteria of having an estimated quality score ≥50 (defined as the estimated completeness of a genome minus five times its estimated contamination)[42] and at least 10× coverage[16,23,43,44].

Three species belonging to *Gammaproteobacteria* containing *pmoA* genes and passing the required criteria for MAG quality (Fig. 2a and Supplementary Data 4) were found exclusively in near-surface sediments (0-10 cmbsf) of the Haima cold seep in the South China Sea (Supplementary Fig. 6). In this surface sediment layer, oxygen is typically still available (dissolved in sediment porewater) owing to its penetration from the overlying oxic water column. Functional

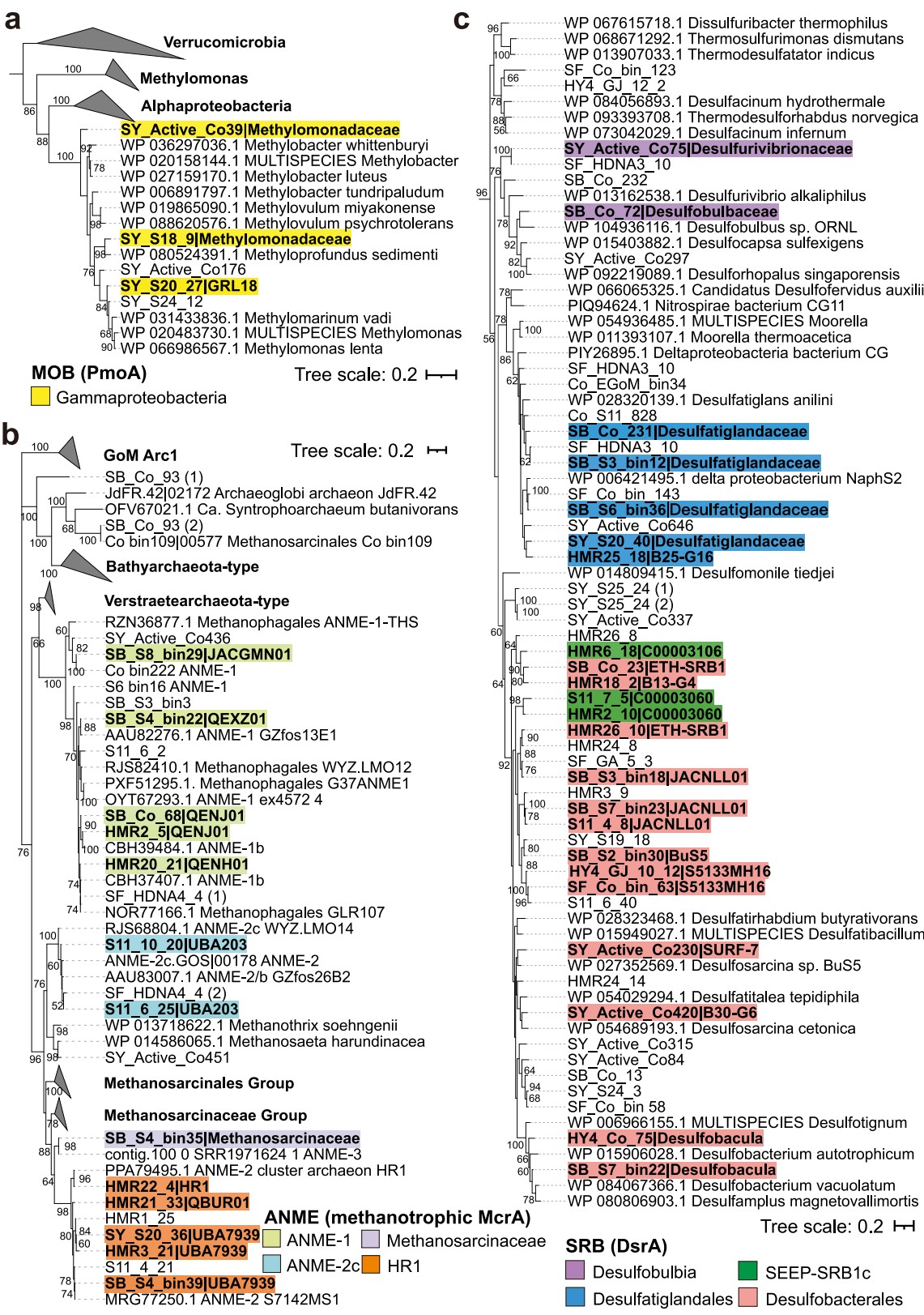

**Fig. 2 | Maximum-likelihood phylogenetic trees of three key functional enzymes in cold seep sediments.** Phylogenetic trees are based on amino acid alignments for **a** PmoA, **b** methanotrophic McrA, and **c** DsrA protein sequences. The sequences from the same taxonomic groups (class, order, or family level) are highlighted in the same colors. The taxonomic information of each sequence is labeled at the family or genus level. Bootstrap values over 50% were shown next to the nodes. Scale bars indicate the average number of substitutions per site. Detailed annotations of species-cluster representative MAGs are provided in Supplementary Data 4.

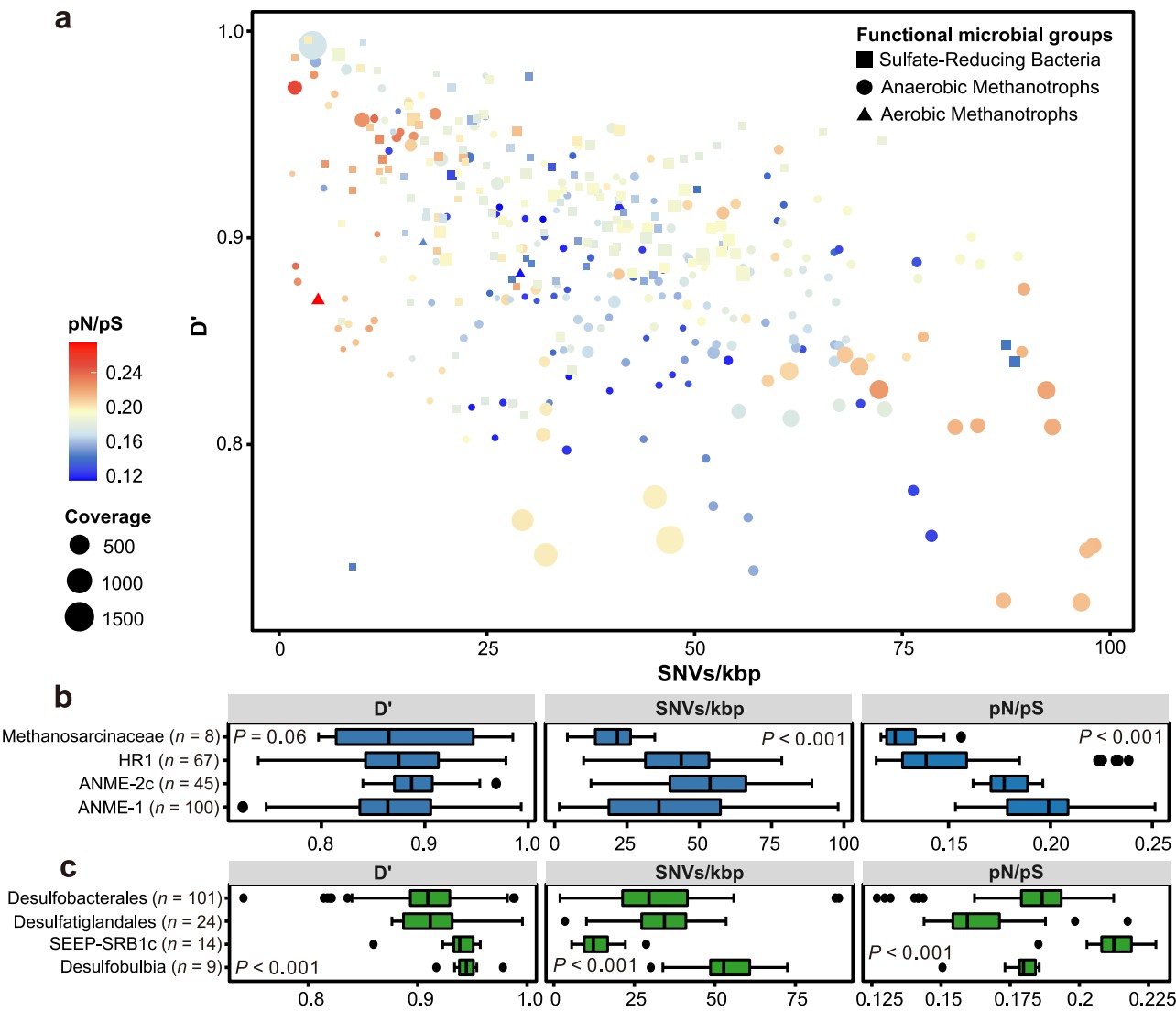

**Fig. 3 | Genome-wide evolutionary metrics of three key functional microbial groups in cold seep sediments. a** Relationships between SNV density (SNVs/kbp; x-axis), linkage disequilibrium (D′; y-axis), the ratio of nonsynonymous to synonymous polymorphisms (pN/pS; symbol color), and genome coverage at the genome level (symbol diameter). Each symbol represents one species-level microbial population, with shapes corresponding to MOB, ANME, or SRB guilds. **b**, **c** Comparison of SNV density, D′ and pN/pS of sulfate-reducing bacteria and anaerobic methanotrophic archaea spanning different taxonomic groups. *P* values of differences across different taxonomic groups were calculated using Kruskal–Wallis rank-sum tests. Boxplot components: center lines, medians; box limits, 25th and 75th percentiles; whiskers, 1.5× interquartile range from the 25th and 75th percentiles; points, outliers. *n* values refer to the number of independent results used to derive statistics. Source data are provided as a Source Data file.

annotations suggest that 13 *Methanosarcinia* and *Syntropharchaeia* species are capable of anaerobic oxidation of methane (Fig. 2b and Supplementary Data 4). These 13 species represent the family-level lineages *Methanosarcinaceae*, HR1, ANME-2c, and ANME-1, and are widely distributed in the majority of the seep sediment samples (53 out of 68) at depths ranging from 1 to 425 cmbsf (Supplementary Fig. 6). A total of 23 bacterial species within the Desulfobacterota have the potential to perform sulfate reduction (Fig. 2c and Supplementary Data 4). These SRB belong to four clades: "C00003060" (aka SEEP-SRB1c[45]), *Desulfobacterales, Desulfobulbia,* and *Desulfatiglandales,* and are widely distributed throughout cold seep sediment columns in 5 locations spanning 1-425 cmbsf (Supplementary Fig. 6). Further phylogenetic analysis (Supplementary Fig. 7) shows that 16 of the SRB species from SEEP-SRB1c and *Desulfobacterales* cluster closely with the typical syntrophic SRB partners of ANME clades (HotSeep-1, Seep-SRB2, Seep-SRB1a, and Seep-SRB1g) reported previously[40,46–48], referred to as syntrophic SRB hereafter.

## Genomic variations across different phylogenetic groups

At the genome level, the 39 species-level MOB, ANME, and SRB identified above were assessed for linkage disequilibrium (D′), the ratio of nonsynonymous to synonymous mutations across the entire genome (pN/pS), the ratio of nucleotide substitutions originating from homologous recombination to those originating from mutation (r/m, gamma/mu; see Methods), and the number of single-nucleotide variations for every thousand base pairs (SNVs/kbp). These evolutionary metrics varied greatly for the populations of MOB, ANME, and SRB (Fig. 3a), showing a wide distribution range for D′ (-0.89 vs 0.72-0.99 vs 0.74-0.99), SNVs/kbp (5-41 vs 2-98 vs 2-89) and pN/pS (0.12-0.27 vs 0.11-0.25 vs 0.13-0.23). D′ values indicate that MOB, ANME, and SRB have not undergone high rates of homologous recombination, similar to the ranges reported for soil bacterial populations across a grassland meadow (D′ values, 0.84-0.96)[18]. Diversity of MOB, ANME and SRB populations, as measured by SNVs/kbp, is not only higher than that observed for soils in grassland meadows (SNVs/kbp, 5-43) using the same SNP calling criteria[18] but also for subseafloor crustal fluids (SNVs/

kbp, 0–65) with a different SNP calling approach[30]. The pN/pS results show that these sedimentary cold seep populations are under purifying selection, suggesting the possibility that these organisms have reached an adaptive optimum in this relatively stable environment, and maintain this by purging nonsynonymous mutations[49]. This result is in line with previous observations reported for bacterioplankton assemblages in sunlit freshwater and marine systems[17,25,50], and microbial populations in other deep-sea biospheres[2,51]. These data (Fig. 3a) also indicate that microbial populations with different functional features in cold seep sediment habitats have diverse evolutionary modes, similar to observations in deep-sea hydrothermal vents with unique attributes[1,28]. Accordingly, statistically significant differences between and within ANME and SRB lineages were observed for almost all of the evolutionary metrics ($P < 0.001$, except for $P = 0.06$ for D′ of ANME groups and $P = 0.04$ for pN/pS between two lineages; Fig. 3b, c and Supplementary Data 6). Additionally, syntrophic SRB partners of ANME (Supplementary Fig. 7) showed lower SNVs/kbp and higher pN/pS values compared to ANME and other SRB groups (Supplementary Fig. 8a).

ANME and SRB populations were further assessed for evolutionary patterns at a finer level of taxonomic resolution (Supplementary Data 5). At the genus or family level, for QENH01, QEXZ01, UBA7939, UBA203, *Methanosarcinaceae*, JACNLL01, and *Desulfatiglandaceae*, negative correlations between D′ and SNVs/kbp reflect a positive relationship between homologous recombination and nucleotide diversity, since lower D′ values mean higher levels of homologous recombination (Fig. 4a and Supplementary Data 6). This is consistent with the positive correlation between r/m and SNVs/kbp for both ANME and SRB (linear regression; $R^2 = 0.34$, $P < 0.001$) (Supplementary Fig. 9a and Supplementary Data 7). The r/m ratio can be used to measure the relative effect of homologous recombination on the genetic diversification of populations[52], and here suggests that ANME and SRB populations preserve high genome-wide diversity via increasing recombination rates to prevent genome-wide selective sweeps brought on by potential environmental changes[15,25]. Negative correlations between SNVs/kbp and pN/pS, as well as D′ for *Methanosarcinaceae* (Fig. 4a, b and Supplementary Data 6), indicate that the population is stabilized by frequent recombination while maintaining a high degree of intrapopulation diversity and an accumulation of synonymous mutations or removal of nonsynonymous mutations. These results point to an ancient divergence within *Methanosarcinaceae* populations[50,53]. In contrast, QENH01 and QEXZ01 from ANME-1, B13-G4 from *Desulfobacterales*, and *Desulfurivibrionaceae* had higher pN/pS values with more single-nucleotide variants (Fig. 4b and Supplementary Data 6). *Desulfurivibrionaceae* populations also possessed high SNVs/kbp and low degrees of within-species recombination (Fig. 4a and Supplementary Fig. 10). These data suggest that *Desulfurivibrionaceae* may be in the process of maintaining nonsynonymous mutations, or subspecies establishment (i.e., speciation)[15]. The SEEP-SRB1c group had the lowest SNVs/kbp, highest pN/pS value, and a low degree of recombination relative to the *Desulfobacterales, Desulfobulbia,* and *Desulfatiglandales* SRB groups (Fig. 3c). Fewer recombination events and lower nucleotide diversity are indicative of genome-wide selective sweeps[25], suggesting that SEEP-SRB1c populations have undergone strong selection.

For populations of UBA203 from ANME-2c, UBA7939 from HR1, JACNLL01, and B13-G4 from *Desulfobacterales*, and *Desulfurivibrionaceae*, the genome coverage (i.e., relative abundances of the populations) and SNVs/kbp fit a positive slope linear regression model (Fig. 4c and Supplementary Data 6), indicating that population size has an important influence on genomic microdiversity[44]. Indeed, SNVs/kbp of ANME was negatively correlated ($r = -0.52$, $P = 0.07$) with archaeal numbers estimated by qPCR of 16S rRNA genes at the SB site[3], whereas a positive correlation ($r = 0.53$, $P = 0.07$) was observed between the SNVs/kbp of SRB and estimated bacterial numbers (Supplementary

Data 8), highlighting the role of population size in impacting how microbial populations evolve[1,28,30]. UBA203 populations with higher abundances were found to show high single-nucleotide variations, which were related to the high mutation rate or accumulation of mutations in the population (Supplementary Fig. 11)[28,54]. Constantly low pN/pS ratios further suggest that nonsynonymous mutations in UBA203 populations might have been purged by purifying selection over a long period[50]. For JACNLL01, high-coverage populations also show relatively high degrees of recombination, but this is not associated with changes in amino acid sequences despite high nucleotide variations (Supplementary Fig. 10)[23]. QENH01 and UBA7939 populations with higher abundances showed higher recombination rates (linear regression; $R^2 = 0.44$ and $R^2 = 0.20$, $P < 0.001$; Fig. 4d and Supplementary Data 6), consistent with the positive linear regression between r/m and coverage (linear regression; $R^2 = 0.81$, $P < 0.001$; Supplementary Fig. 9b and Supplementary Data 7). These findings are similar to the evolutionary success of the marine bacterium SAR11 in the near-surface epipelagic waters of the ocean, where the relationship between population abundance and recombination rate was proposed as the underlying mechanism[50].

## Nucleotide variation of three key functional genes

For *pmoA*, *mcrA*, and *dsrA* genes, SNVs/kbp ranged widely from 0.76 to 123 (45 on average, Fig. 5a). Statistically significant differences between *mcrA* and *dsrA* genes were observed for SNVs/kbp and pN/pS ($P < 0.001$), with *dsrA* genes from typical syntrophic SRB partners of ANME (Supplementary Fig. 7) having higher SNVs/kbp and major allele frequency compared to other SRB groups, whereas pN/pS values were similar (Supplementary Fig. 8b). Based on the pN/pS values (0–1.43, 0.16 on average), *pmoA*, *mcrA*, and *dsrA* genes are under strong purifying selection. The evolutionary fitness of these three key functional genes is consistent with that observed in previous studies for natural comammox *Nitrospira* populations[49] and *Thalassospira* bacterial populations isolated from million-year-old subseafloor sediments[20]. Essential genes and enzymes like these, which catalyze reactions that are difficult to bypass through alternative pathways, are subject to higher purifying selection than less essential ones ($P = 0.17$, Welch two-sample *t*-test, gene vs genome-wide pN/pS)[27,49]. Even though the pN/pS values for the vast majority of functional genes were well below 1 (indicating purifying selection), genes with pN/pS values above 1 and significantly higher than the genomic average are present, suggesting that positive selection is acting on these genes. Although MOB, ANME, and SRB are buried in the seabed where microbial evolution might operate differently than in sunlit habitats[1], the distribution profile of pN/pS is consistent with neutral theory[55] wherein most mutation events are neutral or deleterious[56] (Fig. 5b). Similarly, among rare genes in microbial genomes, evolution was found to proceed largely via neutral processes[57]. On the other hand, studies of the microbial inhabitants of wild bromeliads demonstrate patterns indicating the action of non-neutral processes[58].

For *mcrA*, the three evolutionary metrics (SNVs/kbp, pN/pS, and major allele frequency) were significantly different ($P < 0.001$) among the four ANME groups (Fig. 5c and Supplementary Fig. 12), indicating differences in evolutionary trends. The *mcrA* gene from phylogenetically related QENH01 populations (HMR20_21 and HMR2_5) within the ANME-1 group was found to have high pN/pS values with low synonymous mutation rates, indicating the positive selection or relaxed purifying selection (Supplementary Fig. 13a). For *mcrA* genes from ANME-2c, SNVs/kbp (linear regression; $R^2 = 0.47$, $P < 0.001$) and pN/pS values (linear regression; $R^2 = 0.17$, $P = 0.003$) positively correlated with gene coverage (Supplementary Fig. 14). This suggests that mutations were maintained for *mcrA* genes during the expansion of ANME-2c populations (linear regression; $R^2 = 0.63$, $P < 0.001$).

For *dsrA* genes, SNVs/kbp and major allele frequency exhibited statistically significant differences among the four SRB groups

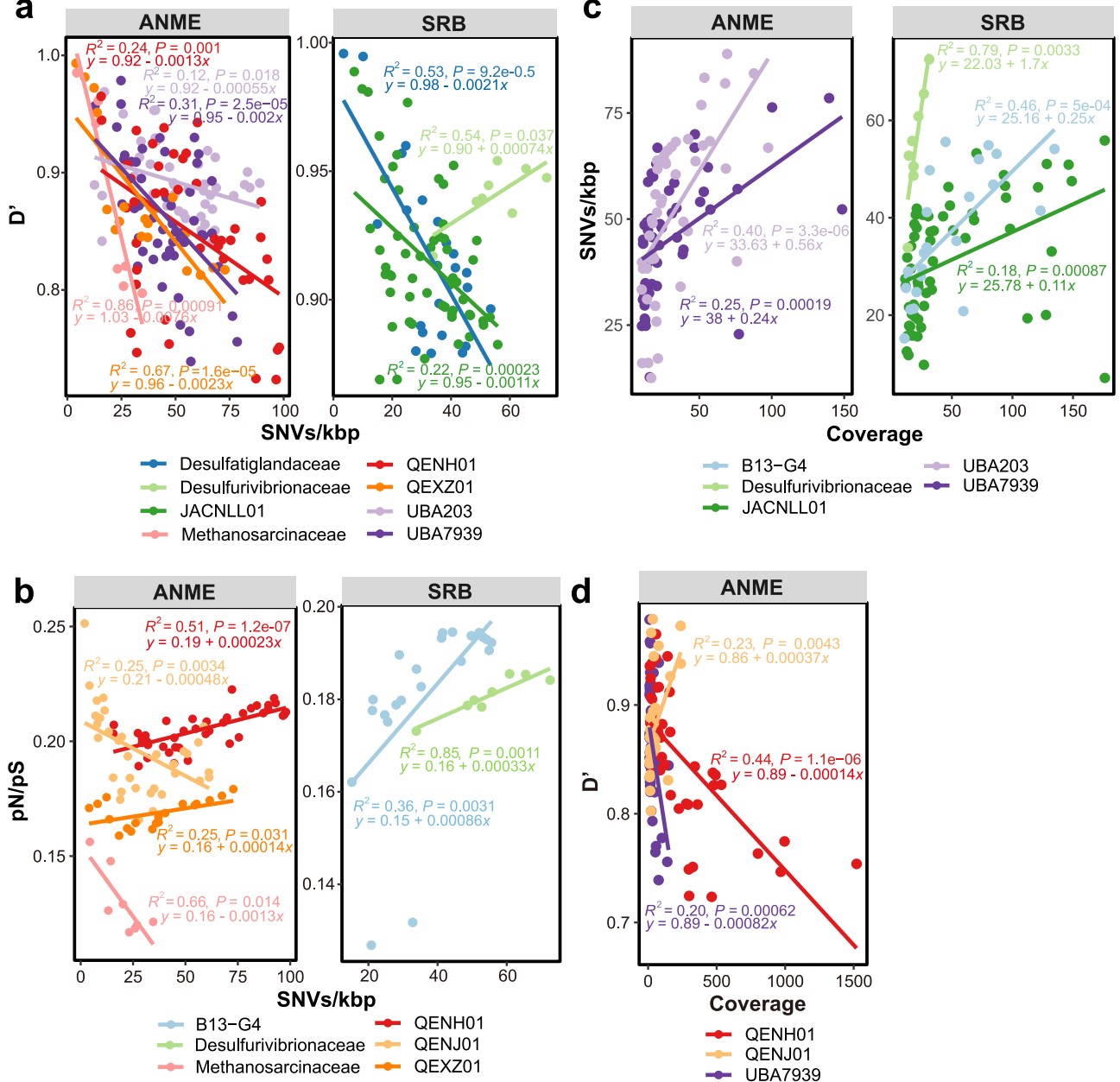

**Fig. 4 | Genome-wide comparison of evolutionary metrics for microbial populations in cold seep sediments. a, b** D' and pN/pS ratio in relation to SNV density. **c, d** SNV density and D' in relation to genome coverage. Each dot represents one species-level microbial population. Linear regressions and $R^2$ values are indicated for different taxonomic groups. Only the significant correlations ($P < 0.05$) among different evolutionary metrics of selected populations are shown. Detailed taxonomic information for analyzed MAGs and statistics for linear regressions are provided in Supplementary Data 5 and 6, respectively.

($P \leq 0.005$), whereas pN/pS values were similar (Fig. 5d). Evolutionary trends (relatively low SNVs/kbp and pN/pS values) were similar in *dsrA* genes from *Desulfobulbia*, SEEP-SRB1c, and *Desulfatiglandales* (Supplementary Fig. 15), indicating these *dsrA* genes to be functionally stable following purifying selection. The *dsrA* gene in the *Desulfobacterales* group also had low pN/pS values (~0.12) but showed a broad range of SNVs/kbp. In one instance, the *dsrA* gene of ETH-SRB1-SB_S7_bin23 populations exhibits abnormally high pN/pS values with fewer synonymous mutations (Supplementary Fig. 13b), indicating that these *dsrA* genes were under strong positive selection and likely further influenced by genetic drift[59].

Major allele frequencies for *mcrA* and *dsrA* genes (0.79 on average) appear to show a direct relationship with SNVs/kbp (Supplementary Figs. 12, 15). High SNVs/kbp correspond to relatively concentrated and high major allele frequencies (~0.8) accompanied by

high coverage, while the distributions of major allele frequency are mostly scattered (0.56–0.99) at low values of SNVs/kbp. Specific major alleles were fixed in most *mcrA* and *dsrA* genes (major allele frequency, 0.70–0.98) with varying degrees of nucleotide diversity (SNVs/kbp, 1–123). For *mcrA* and *dsrA* genes with relatively low major allele frequency (0.56–0.70) and low pN/pS values (0–0.26), genetic heterogeneity is preserved and strongly selected in populations of HR1, SEEP-SRB1c, *Desulfobulbia*, and *Desulfobacterales*, potentially contributing to the diversification of functional genes under different environmental conditions[60].

**Depth- and site-dependent trends of microdiversity**

To determine whether microdiversity within related microbial species is depth-dependent in deep-sea cold seep sediments, the relationship between evolutionary metrics and sediment depth at different cold

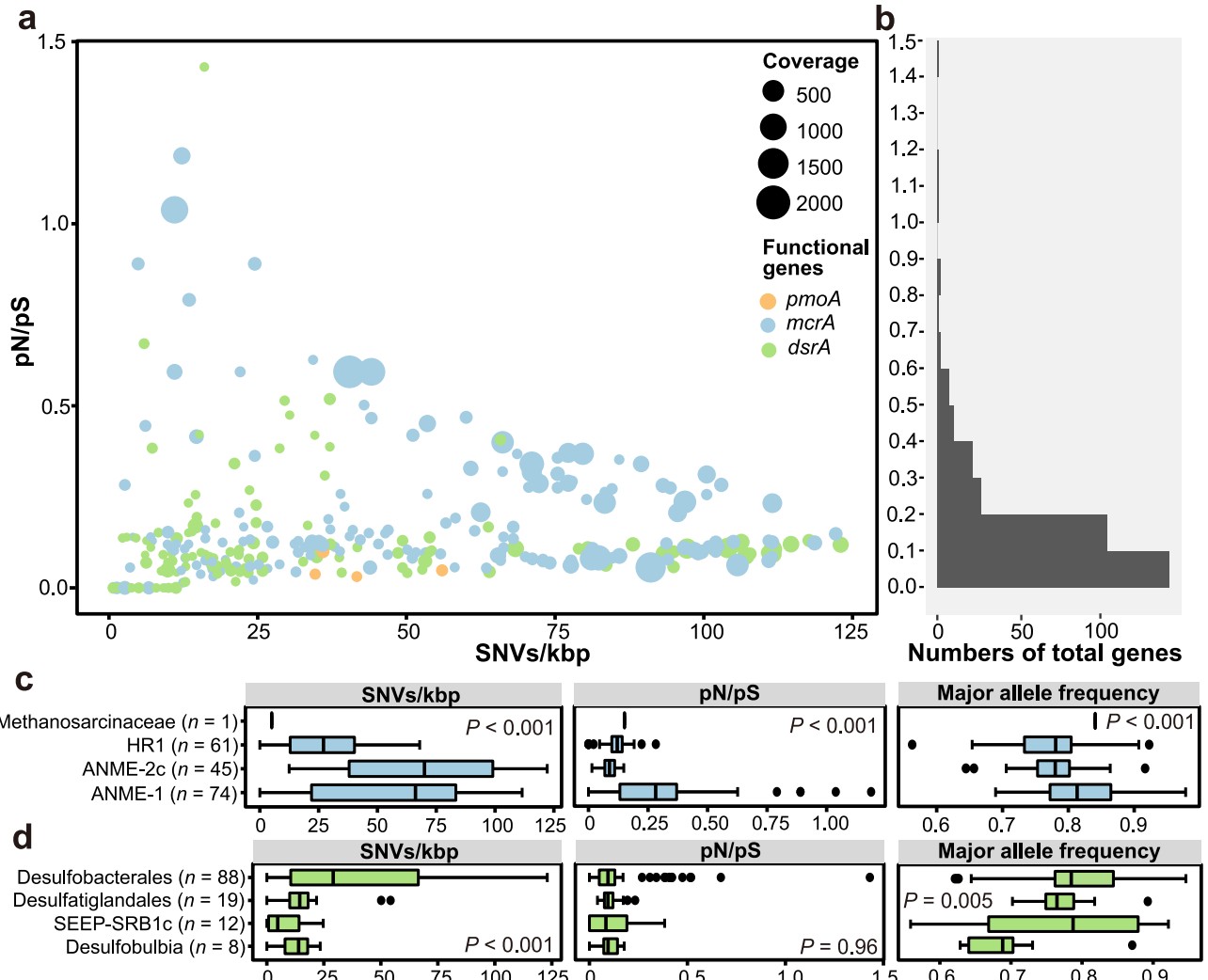

**Fig. 5 | Gene-specific evolutionary metrics of three key functional microbial groups in cold seep sediments. a** Relationships between SNV density, pN/pS, and gene coverage for *pmoA*, *mcrA*, and *dsrA*. Each dot represents one species-level microbial population. **b** Frequency histogram of pN/pS for *pmoA*, *mcrA*, and *dsrA*. **c**, **d** Comparison of SNV density, pN/pS, and major allele frequency of *dsrA* and *mcrA* genes across different taxonomic groups of SRB and ANME, respectively. *P* values of differences across different taxonomic groups were calculated using Kruskal–Wallis rank-sum tests. Boxplot components: center lines, medians; box limits, 25th and 75th percentiles; whiskers, 1.5× interquartile range from the 25th and 75th percentiles; points, outliers. *n* values refer to the number of independent results used to derive statistics. Source data are provided as a Source Data file.

seep sites was examined. At the genome level, SNVs/kbp, pN/pS, and D' showed positive or negative correlations with depth for ANME and SRB populations (Fig. 6a–c and Supplementary Data 9). This suggests that, as depth below the sea floor increases along the sediment column, microbial populations vary in the extent of their microdiversity, homologous recombination, and purifying selection. However, as elaborated below, these trends strongly differed between sites and species, possibly driven by the differences in physiological traits between populations or geochemical conditions between sites. Interestingly, this depth-dependent trend differs from what has been observed in non-seep marine sediments where buried microbial populations show uniformly low genetic heterogeneity regardless of sediment depth[27]. Evolutionary stratification in cold seep sediments is similar to results in a pelagic freshwater environment at Earth's surface where mutation rates were lower deeper in the water column compared to in the surface water of a lake[44]. This is likely related to the higher energy supply and larger population sizes in cold seep sediments compared to most of the sea floor that lacks an influx of high-energy fluids from below[30].

Depth-dependent microdiversity patterns were also observed for *mcrA* and *dsrA* genes. In general, SNVs/kbp and pN/pS are negatively

correlated with sediment depth in most populations, while major allele frequency is correlated positively or negatively with depth in a population- or site-specific manner (Fig. 6d–f and Supplementary Data 9). This suggests that most *mcrA* and *dsrA* genes throughout multiple cold seep sites have lower degrees of microdiversity and are subject to higher levels of purifying selection in ANME and SRB buried deeper in the seabed (i.e., in older sediments). Moreover, pN/pS values of *mcrA* and *dsrA* genes were not significantly different among the depth groups in general (*P* = 0.075 and *P* = 0.55), and could not be solely explained by local abundances of methane or sulfate-metabolizing microorganisms with depth (Supplementary Figs. 6, 16). Additionally, based on observations at genome and gene levels, ANME and SRB populations likely undergo distinct selection pressures arising from sediment depths (Fig. 6).

Cold seep microbiomes are reported to be locally selected and diversified (macrodiversity) by unique benthic biogeochemical conditions and environmental gradients such as for methane and sulfate concentrations[3,10,12,41]. Similarly, the microdiversity of cold seep microbiomes appears to show site-specific patterns (Fig. 6 and Supplementary Data 9). For example, positive correlations were observed between SNVs/kbp, pN/pS, and depth for QENH01 populations from

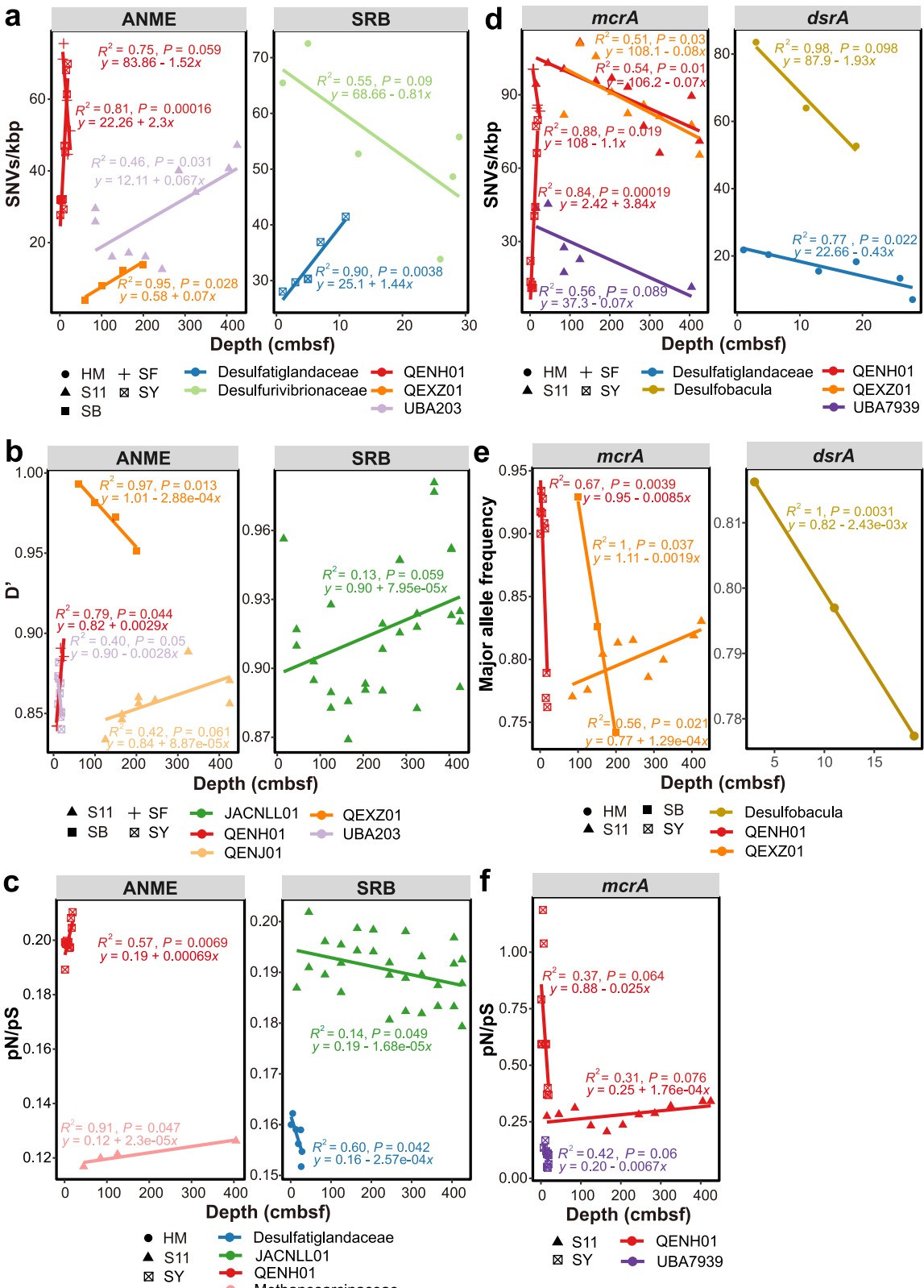

**Fig. 6 | Relationships between evolutionary metrics and depths (cmbsf) of cold seep sediment samples. a–c** Comparison of SNVs/kbp, pN/pS ratio, and D′ against sediment depths at the whole-genome level. **d–f** Comparison of SNVs/kbp, pN/pS ratio, and major allele frequency against sediment depths at the gene level. Each dot represents one species-level microbial population. Linear regressions and $R^2$ values are indicated for different taxonomic groups. Only the significant correlations ($P < 0.1$) between evolutionary metrics and depth of microbial genomes and genes are shown. The shape of the point indicates the cold seep sites. Detailed taxonomic information for analyzed MAGs and statistics for linear regressions are provided in Supplementary Data 5 and 9, respectively.

ANME-1 at the SY site, in contrast to lower microdiversity with depth for this population at the SF site. For QEXZ01 from ANME-1, nucleotide diversity and homologous recombination were positively correlated with depth at the SB site; SNVs/kbp for *Desulfurivibrionaceae* and pN/pS for *Desulfatiglandaceae* were negatively correlated with depth at the HM site, respectively. Nevertheless, the latter two populations exhibited trends of negative coverage with depth at the same site (Supplementary Data 9). Similar trends of microdiversity with depth for these populations were also observed within their *mcrA* and *dsrA* genes at some specific sites (Supplementary Data 9). Since differences between sites depend on seepage dynamics influencing redox zonation[4], evolutionary metrics were also examined in relation to available sediment porewater geochemistry from SY and S11 sites[61,62], including sulfate ($SO_4^{2-}$), dissolved inorganic carbon (DIC) and methane ($CH_4$) levels. Two-tailed Pearson correlation tests (Supplementary Data 10) confirm that evolutionary metrics in several populations were significantly related to these three parameters, with DIC and $CH_4$ likely exerting strong selection pressure on microbial populations based on being negatively correlated with pN/pS values at the genome level.

For three key functional microbial groups on the whole, evolutionary metrics (D′, SNVs/kbp, pN/pS) at the genome level showed significant differences ($P < 0.001$) among different cold seep sites (Supplementary Fig. 17a). This indicates that physicochemical conditions characterizing different cold seep locations also influence intra-population diversity and evolutionary processes. There are also differences in evolutionary metrics (SNVs/kbp and major alleles frequency) for the three functional genes among different cold seep sites ($P < 0.01$) (Supplementary Fig. 17b), indicative of site-dependent microdiversity for functional genes. However, no clear difference in pN/pS was observed between the *pmoA*, *mcrA*, and *dsrA* genes (Supplementary Data 11). This suggests that these key genes from MOB, ANME, and SRB are mostly functionally conserved across different cold seep sites.

## Discussion

Microbial evolution in cold seep sediments is complex and governed by factors that differ from those that influence other parts of the benthos, including energy-limited marine sediments[27], hydrothermal vents[28], and low biomass subseafloor fluids[30]. Methanotrophic and sulfate-reducing populations in cold seep sediments exhibit diverse evolutionary modes with different degrees of nucleotide variation and homologous recombination, suggesting that selection pressure exerted by seeping hydrocarbon fluids enriched in methane influences different populations in different ways. Species of MOB, ANME, and SRB generally undergo low homologous recombination and strong purifying selection, the latter being especially evident in functional genes related to methane (*pmoA* and *mcrA*) and sulfate (*dsrA*) metabolisms. Evolutionary metrics revealed by these genes differed between species but were functionally conserved geographically across different cold seep ecosystems globally, supporting the importance of relatively stable environmental parameters (i.e., continuous supplies of methane and sulfate). Sediment depth not only shapes the community structure of cold seep sediment microbiomes, but is also a determinant of intraspecies microdiversity observed in microbial genes and genomes. Depth-dependent trends of microdiversity thus appear to be a consequence of changing redox conditions and sediment age[5,27]. These observations improve our understanding of principles that drive the evolution of slow-growing deep-sea microbes in one of the most unique subseafloor habitats.

## Methods

### Metagenomes for deep-sea cold seep sediments

Metagenomic data sets were compiled from 68 sediment samples (0 to 430 cmbsf) collected from six globally distributed cold seep sites (Supplementary Fig. 1). These sites are as follows (Supplementary Data 1): Eastern Gulf of Mexico (EGM), Northwestern Gulf of Mexico (GOM-D), Scotian Basin (SB), Haiyang4 (HY4), Site F (SF), and Haima cold seeps in the South China Sea (HM1, HM3, HM5, SY5, SY6, and S11). For samples from the Northwestern Gulf of Mexico, metagenomic data sets along with metadata were downloaded from NCBI Sequencing Read Archive (SRA) and NCBI BioProject databases (https://www.ncbi.nlm.nih.gov/bioproject/PRJNA553005)[63]. The other 63 metagenomic data sets used in this study were obtained from our previous publications[3,11,31,61,62,64,65] (details in Supplementary Data 1). Geochemical parameters ($SO_4^{2-}$, DIC, and $CH_4$) from SY and S11 sites and cell densities from the SB site were collected from our previous publications[3,61,62]. The 68 sediment samples were cataloged into eight groups according to depth below the sea floor (Supplementary Fig. 3): 0–5 cmbsf ($n = 11$); 5–10 cmbsf ($n = 14$); 10–20 cmbsf ($n = 16$); 20–30 cmbsf ($n = 8$); 30–100 cmbsf ($n = 6$); 100–200 cmbsf ($n = 4$); 200–300 cmbsf ($n = 4$); 300–430 cmbsf ($n = 5$).

### Metagenome assembly and binning

Paired-end raw reads were quality-controlled by trimming primers and adapters and filtering out artifacts and low-quality reads using the Read_QC module within the metaWRAP pipeline (v1.3.2; –skip-bmtagger)[66]. Filtered reads from each sample ($n = 68$) were individually assembled and pooled reads from each sampling station ($n = 11$; see Supplementary Data 1) were co-assembled, both using MEGAHIT (v1.1.3; default parameters) based on succinct *de Bruijn* graphs[67]. Contigs less than 1000 bp were removed. For each assembly, contigs were binned using the binning module (parameters: –maxbin2 –concoct –metabat2) and consolidated into a final bin set using the Bin_refinement module (parameters: –c 50 –x 10) within metaWRAP. The quality of the obtained MAGs was estimated by the lineage-specific workflow of CheckM (v1.0.12)[68]. MAGs estimated to be at least 50% complete and with less than 10% contamination were retained.

### Species-level clustering and taxonomic assignment

MAGs reconstructed from individual assemblies and co-assemblies ($n = 79$) were combined and dereplicated for species-level clustering using dRep (v3.2.2)[33] with an average nucleotide identity (ANI) cutoff value of 95%. A total of 1261 MAGs with the highest genome quality from each species cluster were designated as the representative species. The taxonomic classifications of representative MAGs were assigned based on the Genome Taxonomy Database GTDB (release 06-RS202)[37] via the classify workflow of GTDB-Tk (v1.5.1)[69]. To calculate the relative abundance of each MAG, CoverM was used in genome mode (v0.6.0; parameters: –min-read-percent-identity 0.95 –min-read-aligned-percent 0.75 –trim-min 0.10 –trim-max 0.90; https://github.com/wwood/CoverM).

### Functional annotations and phylogenetic analysis

METABOLIC-G, an implementation of METABOLIC (v4.0)[70], was used to predict the metabolic and biogeochemical functional trait profiles of MAGs. For functional genes related to methane and sulfate metabolisms, we also screened the predicted genes against custom protein databases of representative PmoA, McrA, and DsrA sequences (https://doi.org/10.26180/c.5230745) using DIAMOND (v0.9.14), with the threshold criteria of query coverage >80% and percentage identity >50% (McrA and DsrA) or >60% (PmoA). Phylogenetic trees were further constructed to validate the phylogenetic clades of PmoA, McrA (methanogenic or methanotrophic), and DsrA (reductive or oxidative). To build the tree for the three functional proteins, amino acid sequences were aligned using the MUSCLE algorithm[71] included in the software package MEGA X[72]. All positions with less than 95% site coverage were excluded. The maximum-likelihood phylogenetic trees were constructed in MEGA X using the Jones Taylor Thornton matrix-based model, bootstrapped with 50 replicates. To identify typical

syntrophic SRB partners of ANME, a tree was constructed that included 39 genomes from four syntrophic SRB clades (HotSeep-1, Seep-SRB2, Seep-SRB1a, and Seep-SRB1g; see Supplementary Fig. 7) collected from previous studies[40,46-48], along with identified SRB genomes based on DsrA proteins from this study ($n = 23$). The concatenated alignment of 120 single-copy marker genes in bacteria was produced via the identify and align workflow of GTDB-Tk (v1.5.1)[69]. The maximum-likelihood tree was constructed using IQ-TREE (v2.0.5)[73], with the settings: -m MFP -bb. All produced trees were visualized and beautified in the Interactive Tree Of Life (iTOL; v6)[74].

## Calculation of evolutionary metrics

Filtered reads from each sample were mapped to all species-cluster representative MAGs concatenated together using Bowtie2 (v2.2.5; default parameters)[75]. Population statistics and nucleotide metrics, including linkage disequilibrium (D'), nucleotide diversity (SNVs/kbp), nonsynonymous to synonymous mutation ratio (pN/pS), and major allele frequency were calculated from these mappings using the profile module of the inStrain program (v1.5.4; –database mode; default parameters)[16] at genome and gene levels. Genetic annotation of MAGs was performed with Prodigal (v2.6.3; –p meta)[76] for the gene module of inStrain. For SNP calling, the number of quality-filtered reads mapping to the position had to be at least 5× coverage and 5% SNP frequency, while the reads with the variant base had to be above the expected sequencing error rate ($1 \times 10^{-6}$). To estimate the impact of changing sequencing coverage on SNVs/kbp, we analyzed the correlation between SNVs/kbp and the coverage of all genomes. Increased coverage alone did not account for the large differences in SNVs/kbp (Supplementary Fig. 18), which is consistent with previous studies[25,28].

## Inferring recombination rates

Filtered reads were mapped to each of 39 selected representative MAGs using Bowtie2 (v2.2.5; –sensitive-local mode)[75]. The mcorr package (https://github.com/kussell-lab/mcorr)[77] was used to calculate the rate of recombination to mutation (gamma/mu) for each population. MAGs in which (1) normally distributed residuals for the model fit and (2) the bootstrapping mean was within 2× of the final estimate for gamma/mu[18] were retained, resulting in a set of ten genomes for inferring recombination rates.

## Statistical analyses

Statistical analysis was carried out in R (v4.0.0). Shapiro–Wilk and Bartlett's tests were used to assess the normality and variance homogeneity of the data. The Kruskal–Wallis rank-sum test with Chi-square correction was used for comparison of evolutionary metrics in genomes and genes among different groups and sites. Wilcoxon test was used for comparison of evolutionary metrics in genomes and genes of different ANME and SRB groups. Pearson's product-moment correlation was performed to assess the relationship between various evolutionary metrics (D', pN/pS, r/m, coverage, and SNVs/kbp for genomes; pN/pS, major allele frequency, coverage, and SNVs/kbp for genes) and their relationships with sediment depth, cell densities as well as geochemical parameters. Linear regression was used to fit the data and predict the linear correlation between the indexes mentioned above on population. These metrics were used to test the evolutionary processes in the cold seep sediment populations and the effect of sediment depth on them.

## Reporting summary

Further information on research design is available in the Nature Portfolio Reporting Summary linked to this article.

## Data availability

MAGs, files for the phylogenetic trees, and other related information have been uploaded to Figshare (https://doi.org/10.6084/m9.figshare.17195003.v2). MAGs used for the evolutionary analysis have also been deposited in NCBI under BioProject ID PRJNA831433. The databases used in this study include GTDB database R06-RS202 (https://data.gtdb.ecogenomic.org/releases/release202/), KEGG database (http://www.genome.ad.jp/kegg/), TIGRfam (https://tigrfams.jcvi.org/cgi-bin/index.cgi), Pfam (https://www.ebi.ac.uk/interpro/entry/pfam/), custom hidden Markov model (HMM) databases (https://github.com/banfieldlab/metabolic-hmms), dbCAN_seq (https://bcb.unl.edu/dbCAN_seq/), MEROPS (http://merops.sanger.ac.uk/) and the custom protein databases of representative PmoA, McrA, and DsrA sequences (https://doi.org/10.26180/c.5230745). Source data are provided with this paper.

## Code availability

The present study did not generate codes, and mentioned tools used for the data analysis were applied with default parameters unless specified otherwise.

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

## Acknowledgements

The work was supported by the Scientific Research Foundation of the Third Institute of Oceanography, MNR (No. 2022025 to X.D. and No. 2019021 to Z.S.), the National Natural Science Foundation of China (No. 41906076 to X.D.), the Science and Technology Projects in Guangzhou (No. 202102020970 to X.D.), and Guangdong Basic and Applied Basic Research Foundation (No. 2020191024000691 to X.X.). We thank Dr. Xiaoyuan Feng for providing valuable comments.

## Author contributions

X.D., Z.S., and C.R.J.H. designed this study. X.D. and Y.P. performed the analysis. X.D., Y.P., M.W., L.W., W.W., K.J., and C.G. interpreted the data. Y.W., X.X., J.L., and C.R.J.H. contributed to data collection. X.D., Y.P., L.W., and C.R.J.H. wrote the paper, with input from other authors.

## Competing interests

The authors declare no competing interests.
