## [Peer Review File · Nature Communications]

Evolutionary ecology of microbial populations inhabiting deep sea sediments associated with cold seepsREVIEWER COMMENTS

Reviewer #1 (Remarks to the Author):

In "Evolutionary ecology of microbial populations inhabiting deep sea sediments associated with cold seeps," Dong et al evaluate a variety of evolutionary metrics to assess evolutionary patterns in cold seep sediments. The study is interesting and the methods are generally sound. In terms of methodology, one primary concern is that the authors should consider the positive correlation between SNVs/kbp and coverage, because the increasing SNVs/kbp with coverage could be due to a bioinformatic artifact (i.e. as more reads map, it increases the number of variants that can be included in the calculations, thus inflating nucleotide diversity). It is worth discussing in greater detail here.

The manuscript would also benefit from greater clarity about the study goals and motivations, what implications this could have about how these microbes evolve, and what we might expect about cold seep habitats in particular. In light of this, it seems that the most compelling part of the manuscript is the last part—examining how evolutionary trends vary with depth, especially because cold seeps have an energy supply from below, distinguishing them from other subsurface habitats. This could be examined in further detail, especially with regards to how it compares to coverage. One consideration is whether there are cell count numbers accompanying these samples that could be provided for greater context—smaller populations are associated with greater genetic drift and relaxed purifying selection, but populations sizes may not necessarily be smaller in deeper cold seep sediments, as is the case elsewhere in the biosphere.

More specific comments:

Intro

Lines 50-52: can the authors elaborate a bit more on motivations for the study and broader implications?

Lines 90-91: Reference 27 is listed as a preprint, but it is now published in mBio.

Lines 93-95: this section would be strengthened with greater elaboration on a) the biogeochemical importance of these habitats and b) how understanding evolution of these populations would be important. For example, greater elaboration or clarity on the study hypotheses, why you'd expect cold seeps to be different from other regions of the subsurface, i.e. depth-dependent trends?

Methods

Lines 388-389: were bins made from individual assemblies? Or co-assemblies? Please clarify.

Line 419: I see that InStrain (default parameters) was used to calculate SNVs/kbp; could you provide a little bit more information regarding how InStrain calculates SNV density? i.e. what coverage is required at a given site, and how many SNVs within a site must be present to be considered a true SNV? This is particularly helpful because if these results are to be compared with other habitats, especially in the subsurface, it is important to have a sense of whether the calculations were performed similarly. (It looks like InStrain the defaults are minimum coverage of 5x, minimum SNP frequency of 0.05, SNP false discovery rate of 1e-06—is this correct?)

Results

Line 117—what are "species clusters" as used in this manuscript? Bins? MAGs? Or something else?

Line 141: stick to past tense—"were" very abundant?

Lines 148-149- how does this compare with previous cold seep studies, especially 16S/metagenomic data? (There are two papers cited here, so it seems likely this has been seen before, but it would be helpful to say that more explicitly.)

Line 156- note that mcrA genes are used in both methanogenesis and methanotrophy— how did the authors distinguish methanogenic vs methanotrophic mcrA genes? For example, some mcrA genes were found in Methanosarcina (line 168) but the Methanosarcina includes methanogens. The methanogenic/methanotrophic mcrA genes can be distinguished in a phylogenetic tree, and it seems likely the authors used a means to distinguish them (and it's possible I missed something here), but please clarify.

Line 188: is this the average pN/pS across the entire genome? Different ORFs will, of course, have different pN/pS values, so it's important to clarify here.

Line 191-193: any thoughts on why the SNVs/kbp is higher than for other locations? Or is this just due to differences in how SNVs/kbp were calculated?

Lines 227-228: wouldn't this imply the opposite? High pN/pS means maintaining nonsynonymous mutations?

Line 236-237: what does this mean? Please elaborate.

Lines 275-276: what does "ancient mechanisms" mean here/can this sentence be clarified a bit? Are there eukaryotes in this dataset or as a point of comparison in this analysis in some way?

Lies 285-286: is there evidence that increased coverage means that these populations were undergoing a clonal expansion? Are there no other SNPs elsewhere in the MAG? "clonal expansion" has a specific meaning, and so if these populations were indeed undergoing clonal expansions, there should be evidence supporting that.

Lines 298-308: The authors mention that high coverage is associated with high SNVs/kbp (and this is confirmed in Figure 4). This is an important observation that bears further consideration. Was there a general correlation (among all MAGs, for example) between coverage and SNVs/kbp in this study? If so, that could be bioinformatic bias (more reads \diamond more possible SNVs) rather than a pattern arising from for a biological reason, and so this caveat should be discussed and considered in the text.

Lines 310-312: This is another place where being clear about coverage will be important—I does coverage decrease with depth? The decrease in SNVs/kbp with depth is interesting, but it's important to ensure that this is due to biology and not bioinformatics.

Lines 325-333: These findings are very interesting. However, it would be helpful to provide a bit more context here for what metabolisms tend to predominate at shallower and deeper depths, particularly with regards to methanogenesis/methanotrophy and sulfate reduction, to put the pN/pS values for those genes in context.

Figures

Figure 1

The Sankey diagram is nice and conveys useful information, but part B seems less informative and could probably be removed or moved to the supplement to highlight the more useful information in part A.

Figure 2

This is a minor point, but bootstrap values are hard to determine from circle circumferences. Perhaps include bootstrap values over 50 in text next to the nodes?

Figure 4

It's possible I am missing something, but why does panel C only show the SNVs/kbp vs coverage correlation for specific taxa? As mentioned above, this is an important consideration across all taxa, as the change in SNVs/kbp could be due mostly to increased coverage rather than an actual biological trend.

In general here, it's a bit unclear why some of these relationships are shown only for specific taxa—were these the ones that had significant relationships?

Figure 6

This is (in this reviewer's opinion) actually the most interesting result of the manuscript and worth highlighting. However, as noted above, it would be important to examine trends in the coverage of these populations with depth to ensure that the interesting trend observed here isn't due to a bioinformatic artifact.

Reviewer #2 (Remarks to the Author):

In their manuscript entitled "Evolutionary ecology of microbial populations inhabiting deep sea sediments associated with cold seeps" Dong and coworkers describe the assembly and binning of 1261 MAGs from 6 geographically distant cold seep sites. They select 39 of these MAGs, belonging to aerobic and anaerobic methane oxidizing organisms (MOB and ANME) as well as sulfate reducing bacteria (SRB) for evolutionary analysis. The authors use the inStrain software to determine several population statistics and nucleotide metrics based on mapping on the 39 selected genomes, and then use linear regression to compare the selected population statistics between the three clades, and to observed changes with depth in the sediment. While I find the study design of interest, I have several concerns about the interpretation of the results that I would like to see addressed before publication.

general points:

My major concern with this manuscript is the inappropriate use of linear regression analyses on data where the trends are not linear. For example, figure 4a seems to show that a linear correlation can exist between population SNVs per kilobase of sequence, as observed in the methanosarcinaceae (purple). However, this correlation seems to be a function of a single population. In the case of ANME-1 where the authors seemingly lump multiple populations, the data looks like it could represent (at least) 3 distinct linear trends. Whether separation by site or by reference population would be a better strategy to tease apart these correlations is unclear to me, but lumping them at such a high taxonomic level is clearly inappropriate.

Furthermore, the correlations with depth are heavily dominated with by the very small number of deeper sediments. Perhaps it would be good to treat those as a separate category, and see whether trends vary by site or by population. Also, it would be good to see whether there is any indication that the same population (ie the same reference genome) is under different selective pressures at different sites (if closely related populations can be found at different sites, that is). See also my comments on figure 6 below.

In addition, what I am missing in this story is a relation to the geochemical parameters

in the sediment. In some cases, sediment depth can serve as a decent proxy for the environmental conditions, but it seems that the samples used in this study could have come from contrasting types of cold seeps. It would be interesting to see whether geochemical parameters correlate with metrics of selection on the populations.

It is also unclear to me whether the authors have done validation to assess whether their SRB assigned as partners of ANME are indeed partners of ANME? A phylogenomic tree should be able to resolve this. Studying the metrics of population variability between ANME associated SRB and non-ANME-associated SRB in this dataset could also provide interesting insights.

Methods

Although a reference is given, no information is provided about sampling, sample processing DNA extraction, and sequencing methods for the newly sequenced samples (SY5, SY6, S11). Please add this in the methods section, and specify which components were already described in the referenced study.

Please be more specific about the single and co-assemblies done, and how the 39 reference genomes used were generated. As far as I understand, inStrain is quite robust to composite MAGs, as most metrics are concerned, but it would still be good for the reader to understand where the MAGs came from in much greater detail than presented here.

Why was the decision made to only include the MAGs assembled in the present study, and not any other MAGs from the relevant groups? Theoretically, it is possible that there are populations of ANME or SRB present at your study sites for which you did not retrieve a MAG in the binning effort presented in this study. Additionally, it seems to me that certain MAGs did not include the key marker genes that are also used for population metrics in this study (based on the discrepancy between panels in figure 6). This point could have been addressed by using a related MAG with the marker gene.

Was any quality control done for the mapping of the *mcrA* genes? It strikes me as odd that pN/pS for *mcrA* in ANME-1 departs so far from the values for the whole genome, when this is not at all observed at the whole genome level.

Data availability

Figshare is a great repository for tree files and other supporting data, and I thank the authors for supplying those files there. However, it is not appropriate for MAGs. Those should be deposited in and INSDC database (genbank/ENA/DDBJ) so they are easily accessible and searchable by others, and will also be incorporated into derived databases (like the GTDB). Please submit MAGs in the appropriate database before publication.

Both bioproject numbers mentioned in line 378 should also be mentioned in the data availability statement. Please make sure the Bioprojects are public before publication.

Figures:

There is a lot of redundancy in the figures, but also a lot of inconsistency between figures that I can't explain. For example, the number of genes included in the analyses differs per figure panel: figure 5a and 6d show many more *dsrA* sequences than panel 6e, and panel 6d shows many fewer *mcrA* sequences as 6e.

Figure 1

There seem to be some errors in the Sankey diagram in figure 1a. Several lines go from

higher taxonomic ranks directly to the rank of family (e.g. UBA1414 and SZUA-229, but there are others as well). This shouldn't be possible when annotated with the GTDB-tk. Please check the underlying data carefully and correct where needed.

Figure 2

Please specify which taxonomic groups the colors indicate in a legend (preferably on the figure).

Figure 4

In addition to the concerns raised in my general comments above, I wonder what is going on with the linear trendline plotted for the *desulfobulbia*, the points look to be on a straight line, but the trendline does not follow it.

Figure 5

The x axis label of panel A is shifted

Figure 6

The more I look at figure 6, the more confused I get. The taxa included in the plots are not the same across panels, and the taxonomic level at which the data is presented is also not consistent across panels. Please standardize these, and consider splitting the ANME and SRB plots. I also suggest to use the metrics per individual reference genome, and include information on the sampling site (perhaps through shape?) so it becomes clearer what drives the scatter as presented

How do panel B and panel E relate? Why was JACNLL01 assigned to the SRB when the marker gene (*dsrA*) is not present? If this is a consequence of bin incompleteness, would a substitution by a published bin have allowed direct comparison between the whole genome and the gene, as can be done for *mcrA*.

panel e clearly shows at least two populations of ANME1 in the deeper sediments (collected for this study if I'm not mistaken). It strikes me as very interesting to know why one population of ANME-1 is under identical selective pressure as the ANME2c, while the other is not.

REVIEWER COMMENTS

Reviewer #1 (Remarks to the Author):

In “Evolutionary ecology of microbial populations inhabiting deep sea sediments associated with cold seeps,” Dong et al evaluate a variety of evolutionary metrics to assess evolutionary patterns in cold seep sediments. The study is interesting and the methods are generally sound. In terms of methodology, one primary concern is that the authors should consider the positive correlation between SNVs/kbp and coverage, because the increasing SNVs/kbp with coverage could be due to a bioinformatic artifact (i.e. as more reads map, it increases the number of variants that can be included in the calculations, thus inflating nucleotide diversity). It is worth discussing in greater detail here.

Response: We thank the reviewer for detailed and insightful comments. We agree with the reviewer that the correlation between SNVs/kbp and coverage should be considered, and have included this. We found that increased coverage alone did not account for the large differences in SNVs/kbp (Supplementary Figure 18), which is consistent with previous studies (e.g. <https://doi.org/10.1038/s41467-017-01228-6>). Thus, we conclude that our interesting trend observed here isn't due to a bioinformatic artifact. To capture this, the following changes are incorporated into the revision:

Methods (L503-506): “To estimate the impact of changing sequencing coverage on SNVs/kbp, we analyzed the correlation between SNVs/kbp and coverage of all genomes. Increased coverage alone did not account for the large differences in SNVs/kbp (Supplementary Figure 18), which is consistent with previous studies^{25, 28}.”

The manuscript would also benefit from greater clarity about the study goals and motivations, what implications this could have about how these microbes evolve, and what we might expect about cold seep habitats in particular.

Response: We thank the reviewer for this comment. We now added more elaboration of the rationale for our study in the introductory section. Changes made:

Introduction (L52-57): “Cold seeps are widely distributed along continental margins across the globe and serve as highly productive hotspots. Microbial communities in such seeps can be several orders of magnitude more abundant than those in the surrounding marine sediments, potentially impacting how microbial populations evolve over time¹⁻⁵. Cold seeps also possess unique environmental conditions compared to the other benthic environments, shaping local ecology, evolutionary biology, chemistry, and geology^{4, 6-8}.”

Introduction (L97-107): “Cold seep sediments feature supply of hydrocarbon-rich energy and carbon sources, strong redox zonation over depth, abundant electron donors (e.g. sulfide and methane), and frequent fluid exchange^{5, 6, 12}, enabling potential unusual

microbial evolution in these environments. Methane- and sulfur-cycling bacteria and archaea are key players in the microbial communities inhabiting the cold seep sediment ecosystems^{4, 6, 11, 31}. These populations impose strong controls on local chemical and biological regimes through a variety of biogeochemical processes and interactions¹². Despite these dynamics, no direct studies of the evolutionary histories and selection pressures of cold seep sedimentary microorganisms have been conducted, and knowledge of the nucleotide variation of key functional genes related to methane- and sulfur-cycling is lacking.”

In light of this, it seems that the most compelling part of the manuscript is the last part—examining how evolutionary trends vary with depth, especially because cold seeps have an energy supply from below, distinguishing them from other subsurface habitats. This could be examined in further detail, especially with regards to how it compares to coverage.

Response: We appreciate this insightful comment. We now highlight the depth- and site-dependent trends of microdiversity of microbial populations in cold seep sediments and also clarify the depth-dependent trends of microdiversity by site. This includes assessing effects of geochemical factors potentially influencing evolutionary processes. As the reviewer suggests, we now further compare the correlation between coverage and depth for each population. It was found that the coverage of two populations (*Desulfatiglandaceae* and *Desulfurivibrionaceae*) showed negative correlations with sediment depth in one of the sites. Changes made:

Results (L376-388): “Similarly, microdiversity of cold seep microbiomes appears to show site-specific patterns (Figure 6 and Supplementary Table 11). For example, positive correlations were observed between SNVs/kbp, pN/pS and depth for QENH01 populations from ANME-1 at the SY site, in contrast to lower microdiversity with depth for this population at the SF site. For QEXZ01 from ANME-1, nucleotide diversity and homologous recombination were positively correlated with depth at the SB site; SNVs/kbp for *Desulfurivibrionaceae* and pN/pS for *Desulfatiglandaceae* were negatively correlated with depth at the HM site, respectively. Nevertheless, the latter two populations exhibited trends of negative coverage with depth at the same site (Supplementary Table 11). Similar trends of microdiversity with depth for these populations were also observed within their *mcrA* and *dsrA* genes at some specific sites (Supplementary Table 11).”

Results (L388-395): “Since differences between sites depend on seepage dynamics influencing redox zonation⁴, evolutionary metrics were also examined in relation to available sediment porewater geochemistry from SY and S11 sites^{61, 62}, including sulfate (SO_4^{2-}), dissolved inorganic carbon (DIC) and methane (CH_4) levels. Two-tailed Pearson correlation tests (Supplementary Table 12) confirm that evolutionary metrics in several populations were significantly related to these three parameters, with DIC and CH_4 likely exerting strong selection pressure on microbial populations based on being negatively correlated with pN/pS values at genome level.”

One consideration is whether there are cell count numbers accompanying these samples that could be provided for greater context—smaller populations are associated with greater genetic drift and relaxed purifying selection, but populations sizes may not necessarily be smaller in deeper cold seep sediments, as is the case elsewhere in the biosphere.

Response: We thank the reviewer for this comment. In some instances, in the dataset used in this study, bacterial and archaeal numbers were estimated by qPCR of 16S rRNA genes. This allowed this issue to be incorporated into the revised version. SNVs/kbp of ANME is negatively correlated with the numbers of archaea ($r=-0.52$, $P=0.07$), whereas positive correlation between the SNVs/kbp of SRB and bacterial numbers was observed ($r=0.53$, $P=0.07$). As the reviewer suspected, this result highlights the role of population size in impacting how microbial populations evolve. Changes made:

Results (L266-271): “Indeed, SNVs/kbp of ANME was negatively correlated ($r=-0.52$, $P=0.07$) with archaeal numbers estimated by qPCR of 16S rRNA genes at the SB site³, whereas a positive correlation ($r=0.53$, $P=0.07$) was observed between the SNVs/kbp of SRB and estimated bacterial numbers (Supplementary Table 9), highlighting the role of population size in impacting how microbial populations evolve^{1, 28, 30}.”

Methods (L439-441): “Geochemical parameters (SO_4^{2-} , DIC, CH_4) from SY and S11 sites and cell densities from the SB site were collected from our previous publications^{3, 61, 62}.”

More specific comments:

Intro

Lines 50-52: can the authors elaborate a bit more on motivations for the study and broader implications?

Response: We thank the reviewer for this comment. As indicated above, the study objectives are now more clearly elaborated. Changes made:

Introduction (L52-57): “Cold seeps are widely distributed along continental margins across the globe and serve as highly productive hotspots. Microbial communities in such seeps can be several orders of magnitude more abundant than those in the surrounding marine sediments, potentially impacting how microbial populations evolve over time¹⁻⁵. Cold seeps also possess unique environmental conditions compared to the other benthic environments, shaping local ecology, evolutionary biology, chemistry, and geology^{4, 6-8}.”

Lines 90-91: Reference 27 is listed as a preprint, but it is now published in mBio.

Response: We corrected this citation (see Reference 30).

Lines 93-95: this section would be strengthened with greater elaboration on a) the biogeochemical importance of these habitats and b) how understanding evolution of these populations would be important. For example, greater elaboration or clarity on the study hypotheses, why you'd expect cold seeps to be different from other regions of the subsurface, i.e. depth-dependent trends?

Response: We thank the reviewer for this comment. We now added more detail for these aspects in the revised introduction. Changes made:

Introduction (L97-107): “Cold seep sediments feature supply of hydrocarbon-rich energy and carbon sources, strong redox zonation over depth, abundant electron donors (e.g. sulfide and methane), and frequent fluid exchange^{5, 6, 12}, enabling potential unusual microbial evolution in these environments. Methane- and sulfur-cycling bacteria and archaea are key players in the microbial communities inhabiting the cold seep sediment ecosystems^{4, 6, 11, 31}. These populations impose strong controls on local chemical and biological regimes through a variety of biogeochemical processes and interactions¹². Despite these dynamics, no direct studies of the evolutionary histories and selection pressures of cold seep sedimentary microorganisms have been conducted, and knowledge of the nucleotide variation of key functional genes related to methane- and sulfur-cycling is lacking.”

Methods

Lines 388-389: were bins made from individual assemblies? Or co-assemblies? Please clarify.

Response: We appreciate the reviewer flagging this ambiguity and the opportunity to clarify. We used both individual assemblies and co-assemblies for binning process. We changed these sentences for clarity. Changes made:

Methods (L449-452): “Filtered reads from each sample (n = 68) were individually assembled and pooled reads from each sampling station (n = 11; see Supplementary Table 1) were co-assembled, both using MEGAHIT (v1.1.3; default parameters) based on succinct *de Bruijn* graphs⁶⁷.”

Methods (L460-462): “MAGs reconstructed from individual assemblies and co-assemblies (n = 79) were combined and dereplicated for species-level clustering using dRep (v3.2.2)³³ with an average nucleotide identity (ANI) cutoff value of 95%.”

Line 419: I see that InStrain (default parameters) was used to calculate SNVs/kbp; could you provide a little bit more information regarding how InStrain calculates SNV density? i.e. what coverage is required at a given site, and how many SNVs within a site must be present to be considered a true SNV? This is particularly helpful because if these results are to be compared with other habitats, especially in the subsurface, it is important to have a sense of whether the calculations were performed similarly. (It looks like InStrain the defaults are minimum coverage of 5x, minimum SNP frequency of 0.05, SNP false discovery rate of 1e-06—is this correct?)

Response: We thank the reviewer for pointing this out. We now added the sentence to clarify about the calculation of SNVs using inStrain. Changes made:

Methods (L500-503): “For the SNP calling, the number of quality-filtered reads mapping to the position had to be at least 5× coverage and 5% SNP frequency, while the reads with the variant base had to be above the expected sequencing error rate (1×10^{-6}).”

Results

Line 117—what are “species clusters” as used in this manuscript? Bins? MAGs? Or something else?

Response: After binning of metagenomic assemblies, all the final bins were clustered at species-level with an average nucleotide identity (ANI) cutoff value of 95% using dRep (v3.2.2), and the 1261 MAGs of highest genome quality from each species cluster were finally identified as representative species. To clarify “species clusters”, the following changes were made:

Results (L129-132): “After binning of metagenomic assemblies and dereplication of metagenome assembled genomes (MAGs), 1261 species-level clusters (1041 bacteria and 220 archaea; Supplementary Table 2) were recovered according to the suggested threshold of 95% average nucleotide identity (ANI) for delineating species³²⁻³⁴.”

Methods (L460-464): “MAGs reconstructed from individual assemblies and co-assemblies (n = 79) were combined and dereplicated for species-level clustering using dRep (v3.2.2)³³ with an average nucleotide identity (ANI) cutoff value of 95%. A total of 1261 MAGs with the highest genome quality from each species cluster were designated as representative species.”

Line 141: stick to past tense—“were” very abundant?

Response: Changed as suggested.

Lines 148-149- how does this compare with previous cold seep studies, especially 16S/metagenomic data? (There are two papers cited here, so it seems likely this has been seen before, but it would be helpful to say that more explicitly.)

Response: We thank the reviewer for pointing this out. We added a sentence to highlight previously published work. Changes made:

Results (L162-164): “This is in agreement with the recent demonstration that depth-dependent distribution patterns were observed for cold seep microbial communities based on 16S rRNA gene and metagenomic sequencing³.”

Line 156- note that *mcrA* genes are used in both methanogenesis and methanotrophy—how did the authors distinguish methanogenic vs methanotrophic *mcrA* genes? For example, some *mcrA* genes were found in *Methanosarcina* (line 168) but the *Methanosarcina* includes methanogens. The methanogenic/methanotrophic *mcrA* genes can be distinguished in a phylogenetic tree, and it seems likely the authors used a means to distinguish them (and it’s possible I missed something here), but please clarify.

Response: We apologize for the ambiguity here. We used METABOLIC to search the key marker gene *mcrA* of MAGs. We also screened the genes against custom protein databases of representative McrA sequences (<https://doi.org/10.26180/c.5230745>) using DIAMOND (v0.9.14), with the threshold criteria of query coverage >80% and percentage identity >50%. Phylogenetic trees were constructed based on the protein sequences identified from the database and our study. The McrA sequences clustered within the phylogenetic clades of methanotrophic McrA proteins were recognized as the methanotrophic *mcrA* genes. We changed these sentences to be clear. Changes made:

Methods (L471-478): “METABOLIC-G, an implementation of METABOLIC (v4.0)⁷⁰, was used to predict metabolic and biogeochemical functional trait profiles of MAGs. For functional genes related to methane and sulfate metabolisms, we also screened the predicted genes against custom protein databases of representative PmoA, McrA and DsrA sequences (<https://doi.org/10.26180/c.5230745>) using DIAMOND (v0.9.14), with the threshold criteria of query coverage >80% and percentage identity >50% (McrA and DsrA) or >60% (PmoA). Phylogenetic trees were further constructed to validate the phylogenetic clades of PmoA, McrA (methanogenic or methanotrophic) and DsrA (reductive or oxidative).”

Figure legends (L773-774): “Phylogenetic trees are based on amino acid alignments for (a) PmoA, (b) methanotrophic McrA and (c) DsrA protein sequences.”

Line 188: is this the average pN/pS across the entire genome? Different ORFs will, of course, have different pN/pS values, so it’s important to clarify here.

Response: The pN/pS mentioned in section ‘Genomic variations across different

phylogenetic groups' is the average pN/pS across the entire genome. We now clarified the sentence here. Changes made:

Results (L203-204): "...the ratio of non-synonymous to synonymous mutations across the entire genome (pN/pS), ..."

Line 191-193: any thoughts on why the SNVs/kbp is higher than for other locations? Or is this just due to differences in how SNVs/kbp were calculated?

Response: We thank the reviewer for this comment. We compared the calculation methods of SNVs/kbp for different locations. In comparison with a study in grassland meadow conducted by Crits-Christoph et al. (<https://doi.org/10.1038/s41396-020-0655-x>), we found that we had used the same criteria for SNP calling as this study. On the other hand, Anderson et al. (<https://doi.org/10.1128/mbio.00354-22>) called the SNPs of the microbial populations from subseafloor crustal fluids by retaining all entries with > 0 entropy value, SNV positions ≥ 20 coverage in the time point(s) of interest, and if the departure from consensus was $\geq 10\%$, differing from our approach. Differences in how SNVs/kbp are calculated may impact the comparison of SNVs/kbp from different locations, since more SNVs will be retained under the filtering criteria with decreasing coverage. To capture this, we modified the sentence below in the revised manuscript. Changes made:

Results (L213-217): "Diversity of MOB, ANME and SRB populations, as measured by SNVs/kbp, is not only higher than that observed for soils in grassland meadows (SNVs/kbp, 5-43) using the same SNP calling criteria¹⁸ and but also for subseafloor crustal fluids (SNVs/kbp, 0-65) with a different SNP calling approach³⁰."

Lines 227-228: wouldn't this imply the opposite? High pN/pS means maintaining nonsynonymous mutations?

Response: We thank the reviewer for spotting this mistake, which has been corrected. Changes made:

Results (L255-256): "These data suggest that *Desulfurivibrionaceae* may be in the process of maintaining non-synonymous mutations, or subspecies establishment (i.e., speciation¹⁵)."

Line 236-237: what does this mean? Please elaborate.

Response: We thank the reviewer for pointing this out. The positive correlation between the genome coverage and SNVs/kbp showed that the population with large quantity possessed more genomic microdiversity. Now we rephrased this sentence to better clarify this. Changes made:

Results (L265-266): "indicating that population size has an important influence on

genomic microdiversity⁴⁴.”

Lines 275-276: what does “ancient mechanisms” mean here/can this sentence be clarified a bit? Are there eukaryotes in this dataset or as a point of comparison in the analysis in some way?

Response: We thank the reviewer for this comment. We observed the result that the distribution profile of pN/pS values of three key functional genes was consistent with neutral theory, showing that the rate of synonymous substitution was much larger than the nonsynonymous rate. This distribution pattern of pN/pS is usually observed in eukaryotes, and now we also observed this in the genes of prokaryotes. We therefore attribute this phenomenon to an ancient mechanism shared by prokaryotes and eukaryotes. But strictly speaking, there is still a lack of evidence in the inference. To avoid this ambiguity, we deleted the sentence and look forward to other opportunities to dig deeper into this question.

Lies 285-286: is there evidence that increased coverage means that these populations were undergoing a clonal expansion? Are there no other SNPs elsewhere in the MAG? “clonal expansion” has a specific meaning, and so if these populations were indeed undergoing clonal expansions, there should be evidence supporting that.

Response: We thank the reviewer for pointing this out. The highly abundant populations with almost no microdiversity could be recognized as the populations undergoing clonal expansions. We had no evidence that these populations were undergoing a clonal expansion and so in the revision now use just ‘expansion’ instead of ‘clonal expansion’. Changes made:

Results (L319-321): “This suggests that mutations were maintained for *mcrA* genes during the expansion of ANME-2c populations (linear regression; $R^2=0.63$, $P<0.001$).”

Lines 298-308: The authors mention that high coverage is associated with high SNVs/kbp (and this is confirmed in Figure 4). This is an important observation that bears further consideration. Was there a general correlation (among all MAGs, for example) between coverage and SNVs/kbp in this study? If so, that could be bioinformatic bias (more reads □ more possible SNVs) rather than a pattern arising from for a biological reason, and so this caveat should be discussed and considered in the text.

Response: We thank the reviewer for this comment. We agree with the reviewer that the positive correlation between SNVs/kbp and coverage should be considered. To address this concern, we now added the correlation analysis between SNVs/kbp and coverage. We found that increased coverage alone did not account for the large differences in SNVs/kbp (Supplementary Figure 18), which is consistent with previous studies (e.g. <https://doi.org/10.1038/s41467-017-01228-6>). Changes made:

Methods (L503-506): “To estimate the impact of changing sequencing coverage on SNVs/kbp, we analyzed the correlation between SNVs/kbp and coverage of all genomes. Increased coverage alone did not account for the large differences in SNVs/kbp (Supplementary Figure 18), which is consistent with previous studies^{25, 28}.”

Lines 310-312: This is another place where being clear about coverage will be important—I does coverage decrease with depth?

Response: We appreciate this comment and the opportunity to clarify. We took the reviewer’s suggestion and compared the correlation between coverage and depth for each population. We found that the coverage of two populations (*Desulfatiglandaceae* and *Desulfurivibrionaceae*) showed the significantly negative correlation with sediment depth at the HM site. Changes made:

Results (L381-386): “For QEXZ01 from ANME-1, nucleotide diversity and homologous recombination were positively correlated with depth at the SB site; SNVs/kbp for *Desulfurivibrionaceae* and pN/pS for *Desulfatiglandaceae* were negatively correlated with depth at the HM site, respectively. Nevertheless, the latter two populations exhibited trends of negative coverage with depth at the same site (Supplementary Table 11).”

The decrease in SNVs/kbp with depth is interesting, but it’s important to ensure that this is due to biology and not bioinformatics.

Response: We thank the reviewer for this comment. As already indicated above, our correlation analysis between SNVs/kbp and coverage showed that increased coverage alone did not account for the large differences in SNVs/kbp. Changes made:

Methods (L503-506): “To estimate the impact of changing sequencing coverage on SNVs/kbp, we analyzed the correlation between SNVs/kbp and coverage of all genomes. Increased coverage alone did not account for the large differences in SNVs/kbp (Supplementary Figure 18), which is consistent with previous studies^{25, 28}.”

Lines 325-333: These findings are very interesting. However, it would be helpful to provide a bit more context here for what metabolisms tend to predominate at shallower and deeper depths, particularly with regards to methanogenesis/methanotrophy and sulfate reduction, to put the pN/pS values for those genes in context.

Response: We thank the reviewer for this suggestion. In cold seep sediments, the distribution pattern of ANME and SRB is mainly influenced by the concentration of methane and sulfate over the sediment column. SRB are always abundant in shallower depths than in deeper depths, since the sulfate-rich region is in the top sediment column. Different ANME groups (ANME-1, ANME-2, ANME-3) possess different depth-dependent distribution pattern due to their different tolerance levels to various

environmental factors.

To evaluate what metabolisms tend to predominate at shallower and deeper depths, we analyzed the relative abundance of ANME and SRB genomes among the cold seep samples (see Supplementary Figure 6). It was found that only a few of the ANME and SRB genomes showed the significant depth-dependent trend whereas the majority showed no obvious trend. We then catalogued the samples into eight groups according to depth below the seafloor and observed the relative abundance of the three guilds (MOB, ANME and SRB) and corresponding functional genes among these groups (Revised Supplementary Figures 16a-b). We observed that for ANME the relative abundance of genomes ($P=0.32$) and coverage of functional genes were not significantly different among depth groups ($P=0.25$, $P=0.74$), whereas SRB genomes were in lower abundance in deeper sediments compared to shallower depths ($P=0.01$).

Also, we found that the pN/pS values of *mcrA* and *dsrA* genes did not have significant difference among the depth groups in general ($P=0.075$, $P=0.55$; Revised Supplementary Figure 16c). Together with our analysis of depth-dependent trends of microdiversity, the trends of pN/pS values of functional genes varying with depth were population- and site-specific (Revised Figure 6) and thus cannot not be solely explained by local abundances of methane or sulfate-metabolizing microorganisms with depth. Changes made:

Results (L368-371): “Moreover, pN/pS values of *mcrA* and *dsrA* genes were not significantly different among the depth groups in general ($P=0.075$, $P=0.55$), and could not be solely explained by local abundances of methane or sulfate-metabolizing microorganisms with depth (Supplementary Figures 6 and 16).”

Figures

Figure 1

The Sankey diagram is nice and conveys useful information, but part B seems less informative and could probably be removed or moved to the supplement to highlight the more useful information in part A.

Response: Thank the reviewer for this positive comment and suggestion. We removed Figure 1b, which is now in the supplement. See revised Figure 1b and Supplementary Figure 2.

Figure 2

This is a minor point, but bootstrap values are hard to determine from circle circumferences. Perhaps include bootstrap values over 50 in text next to the nodes?

Response: We fully agree with this comment. We now added the bootstrap values over 50 in text in Figure 2, see Revised Figure 2. Changes made:

Figure legends (L775-776): “Bootstrap values over 50% were shown next to the nodes.”

Figure 4

It’s possible I am missing something, but why does panel C only show the SNVs/kbp vs coverage correlation for specific taxa? As mentioned above, this is an important consideration across all taxa, as the change in SNVs/kbp could be due mostly to increased coverage rather than an actual biological trend. In general here, it’s a bit unclear why some of these relationships are shown only for specific taxa—were these the ones that had significant relationships?

Response: Here we only discussed the evolutionary modes of selected populations based on their significant correlation ($P < 0.05$) among different evolutionary metrics. We added such sentences in the figure legend to clarify this. In addition, according to the comments by reviewer #2, we now more explicitly link the evolutionary patterns with microbial populations at finer taxonomic resolutions (see Revised Supplementary Table 6). Also as suggested by the reviewer in the comments above, we added the correlation analysis between SNVs/kbp and coverage. We found that increased coverage alone did not account for the large differences in SNVs/kbp, which is consistent with previous studies (Revised Supplementary Figure 18). Changes made:

Figure legends (L793-796): “Only the significant correlations ($P < 0.05$) among different evolutionary metrics of selected populations are shown. Detailed taxonomic information for analyzed MAGs and statistics for linear regressions are provided in Supplementary Tables 6 and 7, respectively.”

Methods (L503-506): “To estimate the impact of changing sequencing coverage on SNVs/kbp, we analyzed the correlation between SNVs/kbp and coverage of all genomes. Increased coverage alone did not account for the large differences in SNVs/kbp (Supplementary Figure 18), which is consistent with previous studies^{25, 28}.”

Figure 6

This is (in this reviewer’s opinion) actually the most interesting result of the manuscript and worth highlighting.

Response: We thank the reviewer for this comment. We highlighted the depth- and site-dependent trends of microdiversity of microbial populations in cold seep sediment. According to comments by reviewer #2, we clarified the depth-dependent trends of

microdiversity by site and assessed effects of geochemical factors potentially influencing evolutionary processes in this part. Changes made:

Results (L376-388): “Similarly, microdiversity of cold seep microbiomes appears to show site-specific patterns (Figure 6 and Supplementary Table 11). For example, positive correlations were observed between SNVs/kbp, pN/pS and depth for QENH01 populations from ANME-1 at the SY site, in contrast to lower microdiversity with depth for this population at the SF site. For QEXZ01 from ANME-1, nucleotide diversity and homologous recombination were positively correlated with depth at the SB site; SNVs/kbp for *Desulfurivibrionaceae* and pN/pS for *Desulfatiglandaceae* were negatively correlated with depth at the HM site, respectively. Nevertheless, the latter two populations exhibited trends of negative coverage with depth at the same site (Supplementary Table 11). Similar trends of microdiversity with depth for these populations were also observed within their *mcrA* and *dsrA* genes at some specific sites (Supplementary Table 11).”

Results (L388-395): “Since differences between sites depend on seepage dynamics influencing redox zonation⁴, evolutionary metrics were also examined in relation to available sediment porewater geochemistry from SY and S11 sites^{61, 62}, including sulfate (SO₄²⁻), dissolved inorganic carbon (DIC) and methane (CH₄) levels. Two-tailed Pearson correlation tests (Supplementary Table 12) confirm that evolutionary metrics in several populations were significantly related to these three parameters, with DIC and CH₄ likely exerting strong selection pressure on microbial populations based on being negatively correlated with pN/pS values at genome level.”

However, as noted above, it would be important to examine trends in the coverage of these populations with depth to ensure that the interesting trend observed here isn't due to a bioinformatic artifact.

Response: We thank the reviewer for this comment. As suggested by the reviewer, we compared the correlation between coverage and depth for each population. It was found that the coverage of two populations (*Desulfatiglandaceae* and *Desulfurivibrionaceae*) showed the significantly negative correlation with sediment depth at the HM site. Changes made:

Results (L381-386): “For QEXZ01 from ANME-1, nucleotide diversity and homologous recombination were positively correlated with depth at the SB site; SNVs/kbp for *Desulfurivibrionaceae* and pN/pS for *Desulfatiglandaceae* were negatively correlated with depth at the HM site, respectively. Nevertheless, the latter two populations exhibited trends of negative coverage with depth at the same site (Supplementary Table 11).”

Reviewer #2 (Remarks to the Author):

In their manuscript entitled "Evolutionary ecology of microbial populations inhabiting deep sea sediments associated with cold seeps" Dong and coworkers describe the assembly and binning of 1261 MAGs from 6 geographically distant cold seep sites. They select 39 of these MAGs, belonging to aerobic and anaerobic methane oxidizing organisms (MOB and ANME) as well as sulfate reducing bacteria (SRB) for evolutionary analysis. The authors use the inStrain software to determine several population statistics and nucleotide metrics based on mapping on the 39 selected genomes, and then use linear regression to compare the selected population statistics between the three clades, and to observed changes with depth in the sediment. While I find the study design of interest, I have several concerns about the interpretation of the results that I would like to see addressed before publication.

Response: We thank the reviewer for detailed reading and insightful comments. Detailed responses for each comment are provided below.

general points:

My major concern with this manuscript is the inappropriate use of linear regression analyses on data where the trends are not linear. For example, figure 4a seems to show that a linear correlation can exist between population SNVs per kilobase of sequence, as observed in the methanosarcinaceae (purple). However, this correlation seems to be a function of a single population. In the case of ANME-1 where the authors seemingly lump multiple populations, the data looks like it could represent (at least) 3 distinct linear trends. Whether separation by site or by reference population would be a better strategy to tease apart these correlations is unclear to me, but lumping them at such a high taxonomic level is clearly inappropriate.

Response: We thank the reviewer for pointing out this. We agree with the reviewer and reanalyzed these populations at finer taxonomic resolutions to track the trends. Changes made:

Results (L232-235): “ANME and SRB populations were further assessed for evolutionary patterns at a finer level of taxonomic resolution (Supplementary Table 6). At the genus or family level, for QENH01, QEXZ01, UBA7939, UBA203, *Methanosarcinaceae*, JACNLL01 and *Desulfatiglandaceae*, ...”

Results (L250-256): “In contrast, QENH01 and QEXZ01 from ANME-1, B13-G4 from

Desulfobacterales, and *Desulfurivibrionaceae* had higher pN/pS values with more single nucleotide variants (Figure 4b and Supplementary Table 7). *Desulfurivibrionaceae* populations also possessed high SNVs/kbp and low degrees of within-species recombination (Figure 4a and Supplementary Figure 10). These data suggest that *Desulfurivibrionaceae* may be in the process of maintaining non-synonymous mutations, or subspecies establishment (i.e., speciation¹⁵).”

Results (L262-283): “For populations of UBA203 from ANME-2c, UBA7939 from HR1, JACNLL01 and B13-G4 from *Desulfobacterales*, and *Desulfurivibrionaceae*, the genome coverage (i.e. relative abundances of the populations) and SNVs/kbp fit a positive slope linear regression model ... UBA203 populations with higher abundances were found to show high single-nucleotide variations which were related to the high mutation rate or accumulation of mutations in the population (Supplementary Figure 11)^{28, 54}. Constantly low pN/pS ratios further suggest that non-synonymous mutations in UBA203 populations might have been purged by purifying selection over a long period⁵⁰. For JACNLL01, high-coverage populations also show relatively high degrees of recombination, but this is not associated with changes in amino acid sequences despite high nucleotide variations (Supplementary Figure 10)²³. QENH01 and UBA7939 populations with higher abundances showed higher recombination rates ...”

Futhermore, the correlations with depth are heavily dominated with by the very small number of deeper sediments. Perhaps it would be good to treat those as a separate category, and see whether trends vary by site or by population. Also, it would be good to see whether there is any indication that the same population (ie the same reference genome) is under different selective pressures at different sites (if closely related populations can be found at different sites, that is). See also my comments on figure 6 below.

Response: We agree with the reviewer and sincerely appreciate these suggestions. We deleted correlations with sediment depth which are heavily dominated with by the very small number of deeper sediments. We then reanalyzed trends by site and population (see Revised Figure 6). According to the results, we found that the same population is under different selective pressures at different sites. Changes made:

Results (L346-353): “the relationship between evolutionary metrics and sediment depth at different cold seep sites was examined. At the genome level, SNVs/kbp, pN/pS and D’ showed the positive or negative correlations with depth for ANME and SRB populations (Figures 6a-c and Supplementary Table 11). This suggests that, as depth below the sea floor increases along the sediment column, microbial populations vary in the extent of their microdiversity, homologous recombination, and purifying selection. However, as elaborated below, these trends strongly differed between sites and species.”

Results (L362-365): “In general, SNVs/kbp and pN/pS are negatively correlated with sediment depth in most populations, while major allele frequency is correlated positively or negatively with depth in a population- or site-specific manner (Figures 6d-

f and Supplementary Table 11).”

Results (L376-388): “Similarly, microdiversity of cold seep microbiomes appears to show site-specific patterns (Figure 6 and Supplementary Table 11). For example, positive correlations were observed between SNVs/kbp, pN/pS and depth for QENH01 populations from ANME-1 at the SY site, in contrast to lower microdiversity with depth for this population at the SF site. For QEXZ01 from ANME-1, nucleotide diversity and homologous recombination were positively correlated with depth at the SB site; SNVs/kbp for *Desulfurivibrionaceae* and pN/pS for *Desulfatiglandaceae* were negatively correlated with depth at the HM site, respectively. Nevertheless, the latter two populations exhibited trends of negative coverage with depth at the same site (Supplementary Table 11). Similar trends of microdiversity with depth for these populations were also observed within their *mcrA* and *dsrA* genes at some specific sites (Supplementary Table 11).”

In addition, what I am missing in this story is a relation to the geochemical parameters in the sediment. In some cases, sediment depth can serve as a decent proxy for the environmental conditions, but it seems that the samples used in this study could have come from contrasting types of cold seeps. It would be interesting to see whether geochemical parameters correlate with metrics of selection on the populations.

Response: We agree with the reviewer and we were able to collect the geochemical data (SO_4^{2-} , DIC, CH_4) from SY and S11 sites. The correlation analysis between geochemical parameters and evolutionary metrics was performed at the genome and gene levels. The data show that the evolutionary metrics in several populations are significantly correlated with these geochemical parameters. Changes made:

Results (L388-395): “Since differences between sites depend on seepage dynamics influencing redox zonation⁴, evolutionary metrics were also examined in relation to available sediment porewater geochemistry from SY and S11 sites^{61, 62}, including sulfate (SO_4^{2-}), dissolved inorganic carbon (DIC) and methane (CH_4) levels. Two-tailed Pearson correlation tests (Supplementary Table 12) confirm that evolutionary metrics in several populations were significantly related to these three parameters, with DIC and CH_4 likely exerting strong selection pressure on microbial populations based on being negatively correlated with pN/pS values at genome level.”

Methods (L439-441): “Geochemical parameters (SO_4^{2-} , DIC, CH_4) from SY and S11 sites and cell densities from the SB site were collected from our previous publications^{3, 61, 62}.”

It is also unclear to me whether the authors have done validation to assess whether their SRB assigned as partners of ANME are indeed partners of ANME? A phylogenomic tree should be able to resolve this. Studying the metrics of population variability between ANME associated SRB and non-ANME-associated

SRB in this dataset could also provide interesting insights.

Response: We thank the reviewer for this very good suggestion. We built a phylogenomic tree to identify syntrophic SRB partners of ANME and compared the evolutionary metrics of population variability among ANME, syntrophic SRB partners of ANME and other SRB groups (see Revised Supplementary Figures 7 and 8). Changes made:

Results (L196-200): “Further phylogenetic analysis (Supplementary Figure 7) shows that 16 of the SRB species from SEEP-SRB1c and *Desulfobacterales* cluster closely with the typical syntrophic SRB partners of ANME clades (HotSeep-1, Seep-SRB2, Seep-SRB1a and Seep-SRB1g) reported previously^{40, 46-48}, referred to as syntrophic SRB hereafter.”

Results (L229-231): “Additionally, syntrophic SRB partners of ANME (Supplementary Figure 7) showed lower SNVs/kbp and higher pN/pS values compared to ANME and other SRB groups (Supplementary Figure 8a).”

Results (L289-293): “Statistically significant differences between *mcrA* and *dsrA* genes were observed for SNVs/kbp and pN/pS ($P < 0.001$), with *dsrA* genes from typical syntrophic SRB partners of ANME (Supplementary Figure 7) having higher SNVs/kbp and major allele frequency compared to other SRB groups, whereas pN/pS values were similar (Supplementary Figure 8b).”

Methods (L483-491): “To identify typical syntrophic SRB partners of ANME, a tree was constructed that included 39 genomes from four syntrophic SRB clades (HotSeep-1, Seep-SRB2, Seep-SRB1a and Seep-SRB1g; see Supplementary Figure 7) collected from previous studies^{40, 46-48}, along with identified SRB genomes based on DsrA proteins from this study ($n = 23$). The concatenated alignment of 120 single-copy marker genes in bacteria was produced via the identify and align workflow of GTDB-Tk (v1.5.1)⁶⁹. The maximum likelihood tree was constructed using IQ-TREE (v2.0.5)⁷³, with the settings: -m MFP -bb. All produced trees were visualized and beautified in the Interactive Tree Of Life (iTOL; v6)⁷⁴.”

Methods

Although a reference is given, no information is provided about sampling, sample processing DNA extraction, and sequencing methods for the newly sequenced samples (SY5, SY6, S11). Please add this in the methods section, and specify which components were already described in the referenced study.

Response: Sampling, DNA extraction and sequencing methods for SY5, SY6 and S11 have been now reported in our two preprints (<https://doi.org/10.21203/rs.3.rs->

2323106/v1 and <https://doi.org/10.1101/2022.12.21.518016>). The information of metagenomic data was rephrased in the methods section. Changes made:

Methods (L437-439): “Details for other 63 samples were described in our previous publications^{3, 11, 31, 61, 62, 64, 65}, including sampling procedures, DNA extraction and metagenomic sequencing (Supplementary Table 1).”

Please be more specific about the single and co-assemblies done, and how the 39 reference genomes used were generated. As far as I understand, inStrain is quite robust to composite MAGs, as most metrics are concerned, but it would still be good for the reader to understand where the MAGs came from in much greater detail than presented here.

Response: We thank the reviewer for pointing this out. We added more details about the how to generate and choose the 39 reference genomes. Changes made:

Results (L175-179): “A total of 39 MOB, ANME and SRB MAGs were retained as species-cluster representatives for microdiversity analyses, which satisfied threshold criteria of having an estimated quality score ≥ 50 (defined as the estimated completeness of a genome minus five times its estimated contamination)⁴² and at least 10× coverage^{16, 23, 43, 44}.”

Methods (L449-452): “Filtered reads from each sample (n = 68) were individually assembled and pooled reads from each sampling station (n = 11; see Supplementary Table 1) were co-assembled, both using MEGAHIT (v1.1.3; default parameters) based on succinct *de Bruijn* graphs⁶⁷.”

Methods (L460-464): “MAGs reconstructed from individual assemblies and co-assemblies (n = 79) were combined and dereplicated for species-level clustering using dRep (v3.2.2)³³ with an average nucleotide identity (ANI) cutoff value of 95%. A total of 1261 MAGs with the highest genome quality from each species cluster were designated as representative species.”

Methods (L471-478): “METABOLIC-G, an implementation of METABOLIC (v4.0)⁷⁰, was used to predict metabolic and biogeochemical functional trait profiles of MAGs. For functional genes related to methane and sulfate metabolisms, we also screened the predicted genes against custom protein databases of representative PmoA, McrA and DsrA sequences (<https://doi.org/10.26180/c.5230745>) using DIAMOND (v0.9.14), with the threshold criteria of query coverage >80% and percentage identity >50% (McrA and DsrA) or >60% (PmoA). Phylogenetic trees were further constructed to validate the phylogenetic clades of PmoA, McrA (methanogenic or methanotrophic) and DsrA (reductive or oxidative).”

Why was the decision made to only include the MAGs assembled in the present study, and not any other MAGs from the relevant groups? Theoretically, it is

possible that there are populations of ANME or SRB present at your study sites for which you did not retrieve a MAG in the binning effort presented in this study.

Response: To ensure the accuracy of evolutionary analysis, we chose the MAGs which displayed coverage high enough to carry out the microdiversity analyses (at least 10× coverage in our study). To preserve this criterion, only MAGs assembled from our cold seep samples were used to ensure these coverage values in metagenomes. Changes made:

Results (L175-179): “A total of 39 MOB, ANME and SRB MAGs were retained as species-cluster representatives for microdiversity analyses, which satisfied threshold criteria of having an estimated quality score ≥ 50 (defined as the estimated completeness of a genome minus five times its estimated contamination)⁴² and at least 10× coverage^{16, 23, 43, 44}.”

Additionally, it seems to me that certain MAGs did not include the key marker genes that are also used for population metrics in this study (based on the discrepancy between panels in figure 6). This point could have been addressed by using a related MAG with the marker gene.

Response: Sorry for the ambiguity here. The MAGs we used in this study all included the key marker genes. Figure 6 showed the evolutionary pattern of populations at finer taxonomic resolutions. We didn't specify the taxonomic information of these populations and thus caused the misunderstanding. Now we provided the evolutionary analysis and detailed information of these populations (see Revised Figure 4, Figure 6 and Supplementary Table 6). Changes made:

Figure legends (L794-796): “Detailed taxonomic information for analyzed MAGs and statistics for linear regressions are provided in Supplementary Tables 6 and 7, respectively.”

Figure legends (L812-814): “Detailed taxonomic information for analyzed MAGs and statistics for linear regressions are provided in Supplementary Tables 6 and 11, respectively.”

Was any quality control done for the mapping of the *mcrA* genes? It strikes me as odd that pN/pS for *mcrA* in ANME-1 departs so far from the values for the whole genome, when this is not at all observed at the whole genome level.

Response: Quality control was done for the mapping of the *mcrA* genes. The nucleotide metrics of genes were calculated from the mappings using the profile module of the inStrain program. inStrain uses a number of different techniques to handle and reduce read mis-mapping, including competitive mapping, adjusting min_read_ani and MapQ.

The average pN/pS value (0-1.19, 0.31 on average) for *mcrA* in ANME-1 is slightly

higher than the values for the whole genome (0.15-0.25, 0.19 on average). This is mainly due to the abnormal high pN/pS values for *mcrA* in ANME-1, which are mostly from phylogenetically related populations (HMR20_21 and HMR2_5). They were found to have high pN/pS values due to low synonymous mutation rates (pS). Changes made:

Results (L313-317): “The *mcrA* gene from phylogenetically related QENH01 populations (HMR20_21 and HMR2_5) within the ANME-1 group was found to have high pN/pS values with low synonymous mutation rates, indicating the positive selection or relaxed purifying selection (Supplementary Figure 13a).”

Data availability

Figshare is a great repository for tree files and other supporting data, and I thank the authors for supplying those files there. However, it is not appropriate for MAGs. Those should be deposited in and INSDC database (genbank/ENA/DDBJ) so they are easily accessible and searchable by others, and will also be incorporated into derived databases (like the GTDB). Please submit MAGs in the appropriate database before publication.

Both bioproject numbers mentioned in line 378 should also be mentioned in the data availability statement. Please make sure the Bioprojects are public before publication.

Response: We thank the reviewer for pointing this out. We have submitted MAGs in NCBI. Sampling, DNA extraction and sequencing methods for SY5, SY6 and S11 have been now reported in our two preprints (<https://doi.org/10.21203/rs.3.rs-2323106/v1> and <https://doi.org/10.1101/2022.12.21.518016>). These related data will all be made to public before the publication of our manuscript. Changes made:

Methods (L437-439): “Details for other 63 samples were described in our previous publications^{3, 11, 31, 61, 62, 64, 65}, including sampling procedures, DNA extraction and metagenomic sequencing (Supplementary Table 1).”

Data availability (L530-532): “MAGs used for the evolutionary analysis have also been deposited in NCBI under BioProject ID PRJNA831433.”

Figures:

There is a lot of redundancy in the figures, but also a lot of inconsistency between

figures that I can't explain. For example, the number of genes included in the analyses differs per figure panel: figure 5a and 6d show many more *dsrA* sequences than panel 6e, and panel 6d shows many fewer *mcrA* sequences as 6e.

Response: We thank the reviewer for detailed comments. We now modified figures according to the reviewers' comments and provide explanations for the inconsistency in the revised manuscript (see the answers below).

Figure 5a shows all of *mcrA* and *dsrA* sequences in our study. Only the significant correlations ($P < 0.1$) between evolutionary metrics and depth of microbial genomes and genes are shown in Figure 6. Thus, the number of gene sequences shown in Figure 5 and Figure 6d-f is not consistent. To avoid the misunderstanding, we added more explanations in the legend of Figure 6. Changes made:

Figure legends (L810-814): "Only the significant correlations ($P < 0.1$) between evolutionary metrics and depth of microbial genomes and genes are shown. The shape of the point indicates the cold seep sites. Detailed taxonomic information for analyzed MAGs and statistics for linear regressions are provided in Supplementary Tables 6 and 11, respectively."

Figure 1

There seem to be some errors in the Sankey diagram in figure 1a. Several lines go from higher taxonomic ranks directly to the rank of family (e.g. UBA1414 and SZUA-229, but there are others as well). This shouldn't be possible when annotated with the GTDB-tk. Please check the underlying data carefully and correct where needed.

Response: We drew the Sankey diagram using the online website (<https://fbreitwieser.shinyapps.io/pavian/>). It could show a maximum of top 25 taxa with the largest number of MAGs at each level. For example, Gammaproteobacteria (n=98) at class level goes directly to the rank of family SZUA-229 (n=11), and the order level SZUA-229 (n=11) is ignored because the number of order SZUA-229 is not in the top 25 at order level. To avoid the misunderstanding, we added an explanation in the legend of Figure 1. Changes made:

Figure legends (L769-771): "The diagram shows the top 25 taxa with the largest number of MAGs at each level. Detailed statistics for 1261 MAGs are provided in Supplementary Table 2."

Figure 2

Please specify which taxonomic groups the colors indicate in a legend (preferably on the figure).

Response: We thank the reviewer for this comment. We now modified Figure 2 and added the taxonomic groups in the legend (see Revised Figure 2).

Figure 4

In addition to the concerns raised in my general comments above, I wonder what is going on with the linear trendline plotted for the desulfobulbia, the points look to be on a straight line, but the trendline does not follow it.

Response: Sorry for the mistakes in Figure 4. We made a mistake here and now redrew Figure 4 (see Revised Figure 4).

Figure 5

The x axis label of panel A is shifted

Response: Sorry for this error. We now modified it (see Revised Figure 5).

Figure 6

The more I look at figure 6, the more confused I get. The taxa included in the plots are not the same across panels, and the taxonomic level at which the data is presented is also not consistent across panels. Please standardize these, and consider splitting the ANME and SRB plots. I also suggest to use the metrics per individual reference genome, and include information on the sampling site (perhaps through shape?) so it becomes clearer what drives the scatter as presented

Response: We thank the reviewer for this comment. Here we showed the populations which possessed the significant trends of evolutionary metrics varying with depth ($P < 0.1$). To make it clear, as suggested by the reviewer, we split the ANME and SRB plots in Figure 6. We analyzed the populations by site, with sites being shown in shapes in the figure (see Revised Figure 6). We also clarified the taxonomic information of these populations in Revised Supplementary Table 6. Changes made:

Figure legends (L810-814): “Only the significant correlations ($P < 0.1$) between evolutionary metrics and depth of microbial genomes and genes are shown. The shape of the point indicates the cold seep sites. Detailed taxonomic information for analyzed MAGs and statistics for linear regressions are provided in Supplementary Tables 6 and 11, respectively.”

How do panel B and panel E relate?

Response: Panel a-c show the comparison of evolutionary metrics against sediment depths at the whole-genome level, while panel d-f show the comparison at the gene level. Panel B and panel E show the comparison of pN/pS values against sediment

depths of genomes and genes, respectively. To clarify this, we added the headers (ANME, SRB, *mcrA* and *dsrA*) above those graphs in panels a-f (see Revised Figure 6).

Why was JACNLL01 assigned to the SRB when the marker gene (*dsrA*) is not present? If this is a consequence of bin incompleteness, would a substitution by a published bin have allowed direct comparison between the whole genome and the gene, as can be done for *mcrA*.

Response: The genome of JACNLL01 assigned to SRB had the marker gene (*dsrA*). Figure 6 showed the evolutionary pattern of populations at finer taxonomic resolutions. We didn't specify the taxonomic information of these populations and thus caused the misunderstanding. Now we provided the evolutionary analysis and detailed information of these populations (see Revised Figure 4, Figure 6 and Supplementary Table 6). Figure 6 showed the populations which possessed the significant trends ($P < 0.1$) of evolutionary metrics varying with depth. JACNLL01 had significantly negative trends of pN/pS varying with depth but the *dsrA* genes in JACNLL01 didn't show the significant trends. So the comparison of pN/pS values against sediment depths of *dsrA* genes from JACNLL01 was not present in Figure 6e. To avoid the misunderstanding, we added an explanation in the legend of Figure 6. Changes made:

Figure legends (L810-814): "Only the significant correlations ($P < 0.1$) between evolutionary metrics and depth of microbial genomes and genes are shown. The shape of the point indicates the cold seep sites. Detailed taxonomic information for analyzed MAGs and statistics for linear regressions are provided in Supplementary Tables 6 and 11, respectively."

panel e clearly shows at least two populations of ANME1 in the deeper sediments (collected for this study if I'm not mistaken). It strikes me as very interesting to know why one population of ANME-1 is under identical selective pressure as the ANME2c, while the other is not.

Response: We found that the two populations of ANME-1 in the deeper sediments were mostly from QEXZ01 and QENH01 populations. QEXZ01 and ANME-2c had the similar pN/pS values in the deeper sediments, while QENH01 showed relatively high pN/pS values compared to these two populations. The *mcrA* genes with high pN/pS values from QENH01 population belonged to HMR20_21 and HMR2_5 genomes. Changes made:

Results (L313-317): "The *mcrA* gene from phylogenetically related QENH01 populations (HMR20_21 and HMR2_5) within the ANME-1 group was found to have high pN/pS values with low synonymous mutation rates, indicating the positive selection or relaxed purifying selection (Supplementary Figure 13a)."

According to the comment by the reviewer, we modified Figure 6 to show the evolutionary information of populations at finer taxonomic resolutions in each

sampling site. Revised Figure 6 now showed the pN/pS values of *mcrA* genes from QENH01 at S11 and SY sites.

REVIEWERS' COMMENTS

Reviewer #1 (Remarks to the Author):

The reviewers have addressed all of my comments and concerns sufficiently, and I have no further concerns.

Reviewer #2 (Remarks to the Author):

The authors have addressed the methodological concerns I raised well. Thanks for the thorough replies and the changes made to the manuscript.

I would recommend addition of the fine-grained taxonomy used in revised figures 4 and 6 to the trees in figure 2. Not in stead of the broad classification, but added to it, so that the readers have an easy way to see to which major clade each genus level group belongs.

I found the revised interpretation of the results in figure 6 a bit descriptive, (eg line 552 "positive or negative correlations" and several other instances in the following paragraphs) but can understand if a more detailed look at the possible physiological causes underlying the differences between the genera, or between sites for the same genus, is beyond the scope of the manuscript.

REVIEWER COMMENTS

Reviewer #1 (Remarks to the Author):

The reviewers have addressed all of my comments and concerns sufficiently, and I have no further concerns.

Response: We are very pleased to hear this, and we thank the reviewer for constructive comments on the manuscript.

Reviewer #2 (Remarks to the Author):

The authors have addressed the methodological concerns I raised well. Thanks for the thorough replies and the changes made to the manuscript.

Response: We are very pleased to hear this, and we appreciate the reviewer's helpful comments and corrections.

I would recommend addition of the fine-grained taxonomy used in revised figures 4 and 6 to the trees in figure 2. Not in stead of the broad classification, but added to it, so that the readers have an easy way to see to which major clade each genus level group belongs.

Response: We thank the reviewer for this suggestion. Done as suggested (see Revised Figure 2). Changes made:

Legends (L775-777): "The sequences from the same taxonomic groups (class, order or family level) are highlighted in the same colors. The taxonomic information of each sequence is labelled at the family or genus level."

I found the revised interpretation of the results in figure 6 a bit descriptive, (eg line 552 "positive or negative correlations" and several other instances in the following paragraphs) but can understand if a more detailed look at the possible physiological causes underlying the differences between the genera, or between sites for the same genus, is beyond the scope of the manuscript.

Response: We thank the reviewer for this comment and highlighted this point in the manuscript. Changes made:

Results (L343-345): "However, as elaborated below, these trends strongly differed between sites and species, possibly driven by the differences in physiological traits between populations or geochemical conditions between sites."